# A hypomorphic variant in *EYS* detected by genome-wide association study contributes toward retinitis pigmentosa

Koji M. Nishiguchi[1,2,3,22 ✉], Fuyuki Miya [4,5,6,22], Yuka Mori[7], Kosuke Fujita[3,8], Masato Akiyama[9,10], Takashi Kamatani[4,6,11], Yoshito Koyanagi[9,10], Kota Sato [1,12], Toru Takigawa[7], Shinji Ueno[3], Misato Tsugita[1], Hiroshi Kunikata[1], Katarina Cisarova[13], Jo Nishino[4], Akira Murakami [14], Toshiaki Abe[15], Yukihide Momozawa[16], Hiroko Terasaki[3], Yuko Wada[17], Koh-Hei Sonoda[10], Carlo Rivolta [18,19,20], Tatsuhiko Tsunoda [4,5,6], Motokazu Tsujikawa[7], Yasuhiro Ikeda[10,21] & Toru Nakazawa[1,2,8,12]

The genetic basis of Japanese autosomal recessive retinitis pigmentosa (ARRP) remains largely unknown. Herein, we applied a 2-step genome-wide association study (GWAS) in 640 Japanese patients. Meta-GWAS identified three independent peaks at $P < 5.0 \times 10^{-8}$, all within the major ARRP gene *EYS*. Two of the three were each in linkage disequilibrium with a different low frequency variant (allele frequency < 0.05); a known founder Mendelian mutation (c.4957dupA, p.S1653Kfs*2) and a non-synonymous variant (c.2528 G > A, p.G843E) of unknown significance. mRNA harboring c.2528 G > A failed to restore rhodopsin mislocalization induced by morpholino-mediated knockdown of *eys* in zebrafish, consistent with the variant being pathogenic. c.2528 G > A solved an additional 7.0% of Japanese ARRP cases. The third peak was in linkage disequilibrium with a common non-synonymous variant (c.7666 A > T, p.S2556C), possibly representing an unreported disease-susceptibility signal. GWAS successfully unraveled genetic causes of a rare monogenic disorder and identified a high frequency variant potentially linked to development of local genome therapeutics.

---

A list of author affiliations appears at the end of the paper.

Genetic diagnosis of heterogenous inherited disorders became less challenging after next-generation sequencing became widely available. However, although the technological development has substantially improved diagnosis rates, the genetic basis of the disease remains unknown in a large proportion of patients, highlighting the limits of the next-generation sequencing approach. Retinitis pigmentosa, which lacks effective treatment options, is the most common form of inherited retinal degeneration. It is initially characterized by the loss of rod photoreceptors, which mediate night vision, and then involves the loss of cone photoreceptors, which are responsible for daylight vision. Retinitis pigmentosa affects ~1 in 3000 people worldwide. The disease is highly heterogenous presenting with a variety of hereditary patterns[1] ranging from classical Mendelian inheritance to non-Mendelian inheritance due to incomplete penetrance[2,3], hypomorphic allele[4–8], or oligogenecity[4,5,9,10]. However, despite the number of reports of non-Mendelian inheritance causing retinitis pigmentosa, its significance in the context of the overall genetic pathology of the disease is yet to be demonstrated. In Japan, the genetic basis of retinitis pigmentosa remains unknown in up to 70% of cases even after sequencing of all the coding regions and the flanking immediate exon–intron boundaries of the known retinitis pigmentosa genes with a next-generation sequencer (NGS panel test), whole-exome sequencing, or whole-genome sequencing[4,11–13]. There is an urgent need for genetic diagnosis of these unsolved cases, particularly those with the autosomal recessive inheritance pattern of retinitis pigmentosa (ARRP), caused by loss-of-function mutations, because such patients may be amenable to adeno-associated virus (AAV)-mediated gene supplementation therapy[14]. Furthermore, there is a growing interest in the detection of prevalent founder mutations as they may serve as potential candidates of mutation-specific therapy including genome-editing therapy that targets a small specific area of the genome[15–18]. Therefore, its best application would be founder mutations found in a large number of patients. These cases are also candidates for antisense oligo therapy[7,16,19], which allows local treatment of the retinal genome and can target larger genes that cannot be treated with conventional gene supplementation therapy.

Genome-wide association studies (GWAS) are a type of analysis that is most often applied to identify susceptibility loci for common traits, each with a relatively small genetic influence[20]. At the same time, it can also uncover rare variants with strong genetic effects in complex diseases that behave almost as Mendelian alleles in monogenic diseases[21–23]. However, GWAS has never been used to directly search for genetic risks in rare monogenic diseases, and its usefulness in such purposes remains unknown.

By comparing differences in allele frequency in cases and controls, GWAS can provide unbiased means of detecting disease-associated loci evenly across the genome, with a little assumption of the inheritance pattern. This contrasts with case-oriented sequencing approaches, which are often obliged to focus around exons and their boundaries, to identify mutations that follow classic Mendelian inheritance. Thus, GWAS can in theory complement these widely used sequencing approaches by searching for any significant genetic risks that remain undetected.

Here, we report the detection of three disease-associated signals/variants in patients with presumed ARRP (see below), using an integrated approach that combined GWAS with NGS panel test.

## Results

**Detection of the *EYS* locus with GWAS and NGS panel test**. We gathered a total of 944 DNA samples from unrelated Japanese patients who had been diagnosed with retinitis pigmentosa consistent with the autosomal recessive mode of inheritance that typically had at least one affected sibling and no affected members in other generations. In addition, isolated cases with no family history as well as an offspring of consanguineous parents were also included. Control samples comprised 924 Japanese individuals. Most of them had been confirmed to have normal fundus through ocular exam. The cases were mostly from either northeastern or southern Japan, whereas the controls were mostly from the northeast (see "Methods" section for detail). All samples were genotyped with a single-nucleotide polymorphism (SNP) array. To search for undetected genetic risks contributing to ARRP, we carried out a meta-GWAS using two independent data sets. The summary of the data sets and the workflow of the GWAS are summarized in Table 1. Of the 644 cases and 620 controls genotyped in the first GWAS, 432 cases and 603 controls were used for analysis in the end after removing 63 cases and 21 controls that failed quality control (QC) and an additional 149 solved cases in which NGS panel test identified pathogenic mutations to account for the cause of disease[11]. Similarly, after removing 14 cases and 13 controls that failed QC and excluding 78 cases genetically solved with NGS panel test[11], the second GWAS included 208 cases and 287 controls. The results of the two GWASs are summarized in Supplementary Fig. 1, Supplementary Tables 1 and 2. Then these two GWAS data sets were combined (for a total of 640 cases and 890 controls) to carry out a meta-GWAS (Fig. 1a). In this analysis, only the locus encompassing *EYS*, the most frequent ARRP-associated gene in Japanese patients, remained significant (OR = 3.95, $P = 1.18 \times 10^{-13}$). Among signals that did not reach genome-wide significance ($P < 5.0 \times 10^{-8}$), there were 12 other peaks with possible relevance ($P < 1.0 \times 10^{-5}$) in which no known retinitis pigmentosa genes were included (Supplementary Table 3). To investigate whether the *EYS* locus contains multiple variants independently associated with the disease that are not included in the same linkage disequilibrium block, we performed conditional analysis

**Table 1 Number of cases and controls used in 1st and 2nd GWAS.**

| | 1st GWAS | 2nd GWAS | Total |
|---|---|---|---|
| No. of cases/controls genotyped | 644/624 | 300/300 | 944/924 |
| No. of cases/controls that failed QC | 63/21 | 14/13 | – |
| No. of cases/controls after QC | 581/603 | 286/287 | 867/890 |
| No. of cases solved by sequencing | 149 | 78 | – |
| No. of cases/controls used in GWAS | 432/603 | 208/287 | 640/890 |
| No. of genotyped SNV used | 523,187 | 522,207 | – |
| No. of SNV after imputation | 10,673,864 | 10,383,808 | – |
| $\lambda$ | 1.061 | 1.037 | – |

*GWAS genome-wide association study, No. number, QC quality control, SNV single-nucleotide variant.*

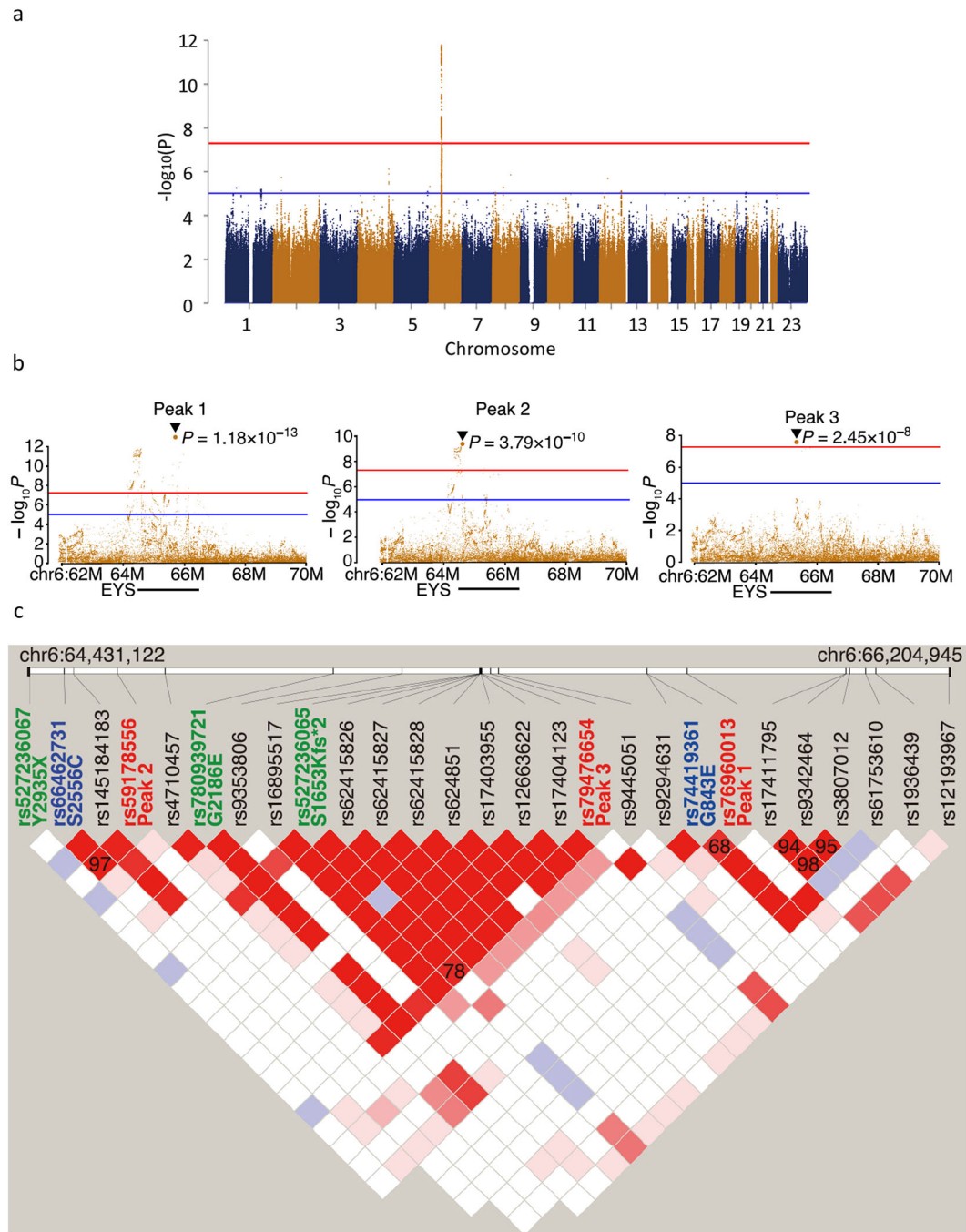

**Fig. 1 Genome-wide association study (GWAS) of ARRP patients and detection of three independent signals in EYS. a** Results of a meta-GWAS displayed as a Manhattan plot. Genome-wide significance ($P = 5.0 \times 10^{-8}$) and possible significance ($P = 1.0 \times 10^{-5}$) are marked with red and blue lines, respectively. A single peak at the *EYS* locus surpassed genome-wide significance. **b** Results of a conditional analysis presented as a regional plot. Three independent peaks at $P < 5.0 \times 10^{-8}$ were delineated after conditioning (Peaks 1–3). **c** Linkage disequilibrium plot using all non-synonymous variants (identified in >5% of cases) and lead variants for Peaks 1–3 identified in GWAS in presumed ARRP patients. The linkage disequilibrium plot was generated using Haploview (ver. 4.1). The default color setting of the software was used for block color setting (D'/LOD). The numbers on the blocks indicate $r^2 \times 100$; numbers are shown on the blocks only for pairs with $r^2 > 0.3$. Peaks 1, 2, and 3 were in linkage disequilibrium with G843E, S2556C, and S1635Kfs, respectively. The lead variants for Peaks 1–3 are shown in red. Reported pathogenic founder mutations[11] are shown in green, while non-synonymous variants linked to the lead variants are shown in blue. Note, S1653Kfs, a reported founder mutation linked to a GWAS lead variant, was shown in green.

for the *EYS* locus. The subsequent conditional analysis revealed 3 independent genome-wide significant signals ($P < 5.0 \times 10^{-8}$; Peaks 1–3 in the order of significance, Table 2 and Fig. 1b) in the *EYS* locus. We then checked the associations of the most significant variants (lead variants), tagging each signal and non-synonymous and splice site variants in the 640 cases using the

NGS panel test results[11]. While Peak 1 and Peak 3 were each linked to a low allele frequency variant (allele frequency < 0.05), Peak 2 was associated with a common non-synonymous variant. More specifically, Peak 1 (rs76960013, allele frequency = 0.0414, odds ratio = 3.95, $P = 1.18 \times 10^{-13}$) was in linkage disequilibrium ($r^2 = 0.68$) with c.2528 G > A (p.G843E; hereafter

**Table 2 Peaks detected with conditional analysis of the *EYS* locus and associated non-synonymous variants.**

|  | Peak 1 | Peak 2 | Peak 3 |
|---|---|---|---|
| dbSNP rsID | rs76960013 | rs59178556 | rs79476654 |
| Gene info | EYS:intron | EYS:intron | EYS:intron |
| Ref/Alt | A/G | C/A | T/C |
| ToMMo AF | 0.0414 | 0.2161 | 0.0005 |
| Odds ratio | 3.95 | 1.83 | 16.46 |
| *P*-value | 1.18E−13 | 3.79E−10 | 2.45E−08 |
| Linked variant | G843E | S2556C | S1653Kfs |
| dbSNP rsID | rs74419361 | rs66462731 | rs527236065 |
| ToMMo AF | 0.0171 | 0.214 | 0.0044 |
| Odds ratio | 3.75 | 1.79 | NA† |
| *P*-value | 1.06E−09 | 1.56E−09* | NA† |
| Corrleation ($r^2$) | 0.68 | 0.97 | 0.78 |
| PolyPhen-2 | Damaging | Possibly damaging | No data |
| SIFT | Damaging | Damaging/ tolerated | Damaging |
| MutationTaster | Disease causing | Polymorphism | Disease causing |
| CADD score | 23.8 | 22.9 | No data |

Information on the three independent peaks detected in this study and exonic variants in linkage disequilibrium are presented.
*P-value for S2556C was calculated after conditional analysis.
†The odds ratio and P-value for S1653Kfs was not available (NA), because the variant was not included in the imputed genotypes of the GWAS analysis.

termed G843E; Table 2). G843E with an allele frequency (0.0171) unusually high for ARRP has been described in conflicting ways in past reports, as having uncertain significance[23], being non-pathogenic[24], possibly being pathogenic (although without sufficient supporting evidence)[25], and was unreported in the two largest genetic screening projects targeting Japanese retinitis pigmentosa patients[9,11]. The second peak, with much higher allele frequency and lower odds ratio (Peak 2; rs59178556, allele frequency = 0.2161, odds ratio = 1.83, $P = 3.79 \times 10^{-10}$), was in strong linkage disequilibrium ($r^2 = 0.97$) with a common non-synonymous variant, i.e., c.7666 A>T (p.S2556C; hereafter termed S2556C variant; Table 2) registered as benign/likely benign in the ClinVar (https://www.ncbi.nlm.nih.gov/clinvar/). Peak 3 (rs79476654, allele frequency = 0.0005, odds ratio = 16.46, $P = 2.45 \times 10^{-8}$) was in linkage disequilibrium ($r^2 = 0.78$) with c.4957dupA (p.S1653Kfs*2) (hereafter termed S1653Kfs; Table 2), recognized as a founder autosomal recessive mutation[11]. It remained statistically significant even after removing solved cases with biallelic *EYS* mutations (including homozygotes and compound heterozygotes with S1653Kfs) screened by NGS panel test prior to GWAS because a large number of heterozygous carriers of S1653Kfs mutation remained genetically unsolved[9,11,24]. Haplotype analysis of *EYS* based on SNP array data and the results of NGS panel test in retinitis pigmentosa patients confirmed that none of the lead variants of the identified signals were in linkage disequilibrium with c.C8805A (p.Y2935X) or c.G6557A (p.G2186E) (hereafter termed G2186E), the two other known founder mutations in this gene[9], in contrast to S1653Kfs, which was in linkage disequilibrium with Peak 3 (Fig. 1c, Table 2 and Supplementary Table 4). This suggests that Peaks 1 and 2 represent under-recognized genetic risks in *EYS*. Thus, GWAS in combination with the NGS panel test successfully detected disease-associated variants overlooked by simple sequence-based approaches in a rare monogenic disease.

**Expression analysis of G843E allele in genome-edited patient-derived lymphoblasts**. The allele frequency (0.2140) of S2556C, linked to Peak 2, was undoubtedly too high for a pathogenic

Mendelian mutation causing a rare monogenic disease. On the other hand, the allele frequency of G843E linked to Peak 1 was much lower (0.017), yet still too high for a classical AR allele, raising the possibility that it represents a reduced penetrant or hypomorphic ARRP allele. Meanwhile, it is also possible that a true ARRP mutation in linkage disequilibrium with Peak 1 exists deep in the non-coding region. However, the vast majority of the known pathogenic mutations in *EYS* are either nonsense, frameshift, or splice site mutations[11] that would presumably result in a qualitative alteration in the mRNA sequence. Thus, we directed our search to the main variants that could affect the coding sequence. For this purpose, we carried out two experiments. First, we performed whole-genome sequencing (WGS) in two G843E homozygotes and two compound heterozygotes (G843E and S1653Kfs or G2186E). We found that there were no obvious structural variants in *EYS* that affected the coding sequence. A splice site prediction analysis[24] also detected no coding and non-coding variants that could alter splicing in these patients. Second, we established patient-derived lymphoblasts (or lymphoblast cell line; LCL) from homozygotes of G843E and S1653Kfs and studied the expression of *EYS* mRNA by forced transcription of *EYS* through the insertion of a constitutively active *CAG* promoter immediately upstream of the initiation codon of the gene (Supplementary Fig. 3a, b)[15]. Among the seven main transcript variants reported for *EYS*[25], the retina-specific long isoforms (transcription variants 1 and 4) are considered essential for photoreceptor biology[25,26]. RT-PCR followed by Sanger sequencing indicated that mRNA containing G843E (exon 16) was expressed without the loss of the C-terminal end of the retina-specific long isoform (Fig. 2a–c). This was unlike transcripts with homozygous S1653Kfs that resulted in the loss of the long isoform via nonsense-mediated decay, which was successfully rescued by replacing the mutation with wild-type sequence through genome editing (Fig. 2d). These results argue against the presence of an intronic mutation in linkage disequilibrium with Peak 1 that results in altered splicing and a presumed premature termination of the reading frame, but support G843E as the causal mutation linked to Peak 1. Meanwhile, the presence of a non-coding variant that causes the disease through changes in the regulation of *EYS* transcription cannot be ruled out.

**Functional analysis of *EYS* G843E in zebrafish**. *EYS* gene is absent in mammalian laboratory animals such as mice, rats, or rabbits. And zebrafish (*Danio rerio*) is the only model in which loss-of-function mutations in the homologous *eys* have been shown to recapitulate photoreceptor degeneration observed in retinitis pigmentosa patients with *EYS* mutations[27–29]. Importantly, G843 residing in the Epidermal Growth Factor-like domain is conserved across various species ranging from zebrafish, chicken, and zebrafish (Fig. 3) As an expression of G843E mutant in the mRNA has been confirmed with the patient-derived LCL (Fig. 2), we used zebrafish to directly assess the function of the mutant allele. Endogenous Eys protein localized near the basal interface of the connecting cilium of the photoreceptors in adult fish (Fig. 4a, b). During development, Eys expression was observed after 4 days post-fertiliztion (dpf, Fig. 4c–f). There was some staining in the inner nuclear, inner plexiform layer, and ganglion cell layer, which was in line with that reported in human[30]. However, the role of Eys outside of photoreceptors is unknown.

To perform specific and effective knockdown of eys, we prepared three different splice site morpholinos (SPMO1-3) and compared their effects on eys expression. SPMOs sometimes yield abnormal splicing variants. However, all three SPMOs did not show evidence of such abnormalities (Fig. 4g), which allowed us

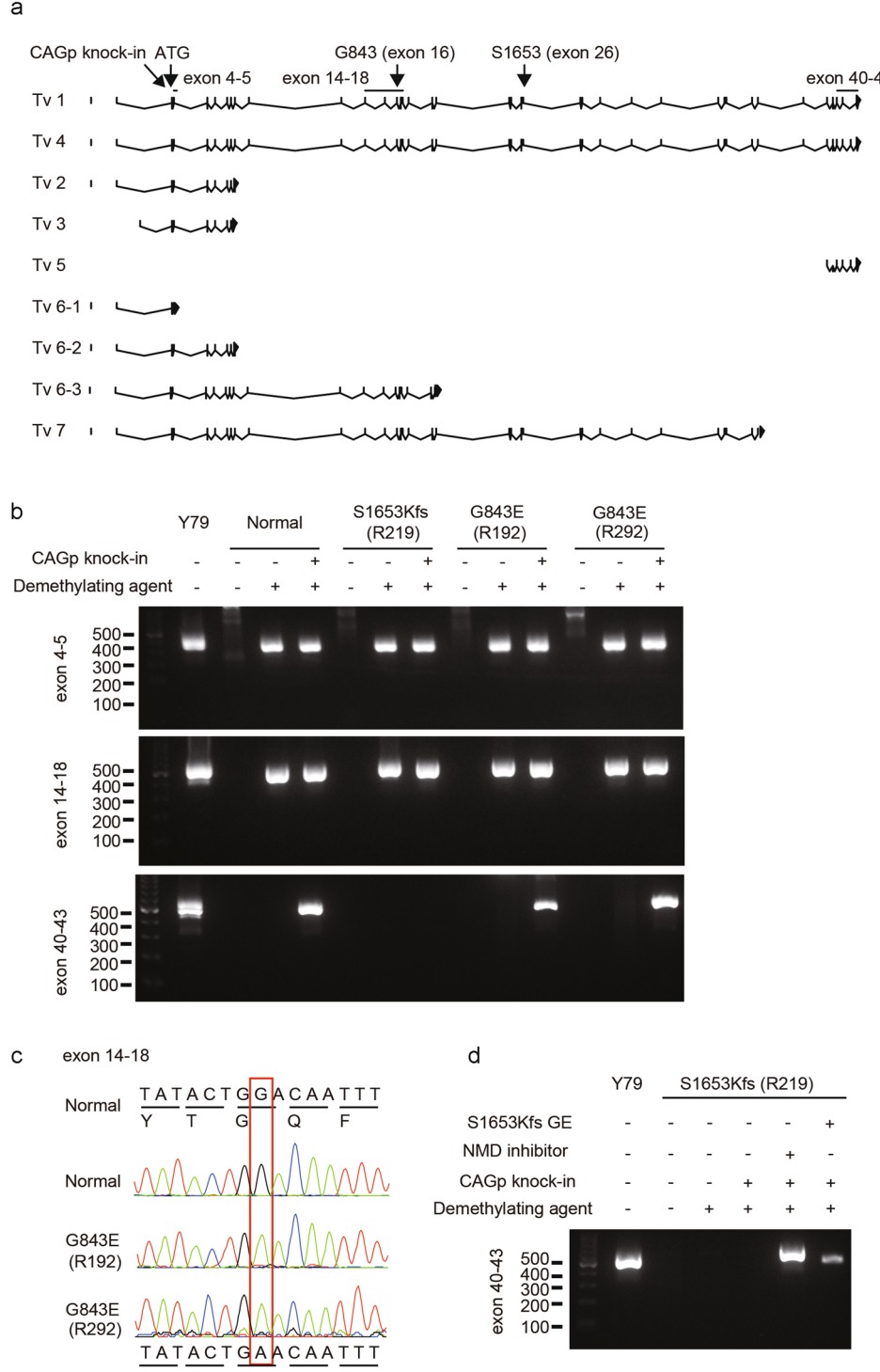

**Fig. 2 Degradation of the *EYS* G843E mutant mRNA in patient-derived lymphoblastoid cell lines (LCLs). a** A schematic map of the RT-PCR primer designed in relation to the exon–intron structure and mutations (G843E and S1653Kfs) in *EYS* and published transcript variants (Tv)[25]. The locations of G843E (Exon 16) and S1653Kfs (Exon 26) are indicated by the arrows. Exon numbers are based on Tv1. Note, Tv5 was identified only in fibroblasts[25]. **b** RT-PCR analysis. The regions for exons 5–6, exons 14–18, and exons 40–43 of *EYS* were amplified on cDNA generated from patient-derived lymphoblast cell lines with wild-type *EYS* (normal), homozygous S1653Kfs, and homozygous G843E. The Y79 retinoblastoma cell line was used as a positive control. Note C-terminal exons of the long isoform Tv1 were detected in LCLs with homozygous G843E but not in that with homozygous S1653Kfs. Sanger sequencing of the RT-PCR amplicon confirmed the expression of the G843E mutation using a primer pair targeting exons 14–18. Meanwhile, mRNA for exons 4–5 and 14–18 were detectable, possibly reflecting the differential expression of distinct *EYS* isoforms[25]. **c** Chromatogram for RT-PCR amplicon (exons 14–18). Note, G843E variant is present in the patient's mRNA. **d** RT-PCR analysis after mutation replacement genome-editing treatment (GE) or inhibition of nonsense-mediated mRNA decay (NMD) in LCL from an S1653Kfs homozygote, after which expression of exons 40–43 was detected.

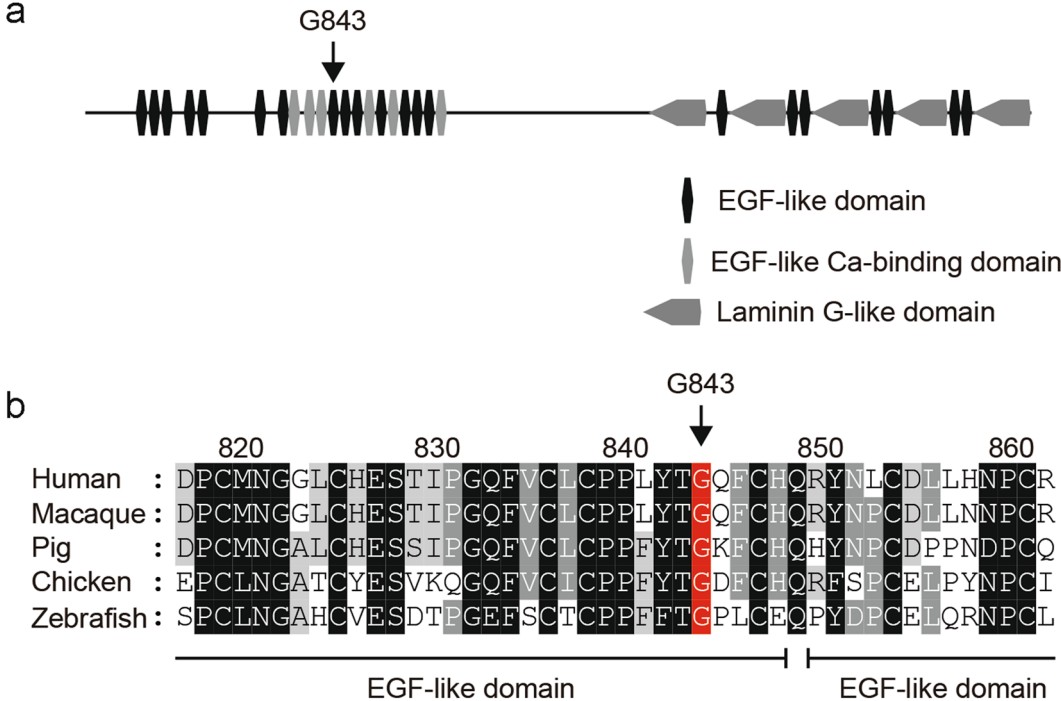

**Fig. 3 Domain structure of *EYS* and conservation of G843 across diverse species. a** Domain structure of *EYS* in relation to G843E. **b** Conservation of G843 across diverse species ranging from zebrafish to humans. The multiple sequence alignment was generated using ClustalW (https://clustalw.ddbj. nig.ac.jp/). Accession numbers of the protein sequences used for sequence comparison are as follows: human, NM_001142800.1; macaque, XM_011737495.1; pig, XM_021084496; chicken, XM_015284845.1; zebrafish, XM_009307513.

to assess their knockdown efficacy by quantitative RT-PCR analyses. All three SPMOs showed suppression of *eys* mRNA (range, 58–88%; Fig. 4h, Supplementary Data 1). However, as SPMO1 appeared least effective, it was co-injected with ATG-MO targeting the first methionine for downstream experiments. Using these morpholinos we examined the mislocalization of rhodopsin. Rhodopsin mislocalization has been reported in mice models of RP[31–33]. In addition, as eys is localized to the connecting cilium, the mislocalization of rhodopsin caused by defective ciliary transport is a reasonable phenotype. Moreover, we have demonstrated that the mislocalization can cause photoreceptor cell death in zebrafish[34]. Furthermore, a zygotic mutant (knockout of *eys* in zebrafish) also shows the same phenotype[29]. The injection of these different morpholinos (SPMO1 + ATG-MO, SPMO2, and SPMO3)-induced mislocalization of rhodopsin in photoreceptors (Fig. 4i–l). This precedes photoreceptor death and is the expected phenotype for eys knockdown zebrafish.

More importantly, this phenotype observed also at 7 dpf following injection of SPMO1 + ATGMO, or SPMO2 (Fig. 4m, n) was rescued by co-injection of human wild-type EYS mRNA (Fig. 4p). On the other hand, the rescue effect was significantly diminished when SPMO1 + ATGMO, or SPMO2 and EYS mRNA with G843E variant were co-injected (Fig. 4o, q and Supplementary Data 1). These results provide direct evidence for the dysfunction of *EYS* caused by G843E.

**Enrichment of G843E in genetically unsolved heterozygous carriers of another *EYS* mutation.** A recent large-scale mutation screening project in 1,204 Japanese retinitis pigmentosa cases revealed an unusually high frequency of carriers of heterozygous deleterious mutations in *EYS*, accounting for 25.1% of the unsolved cases[11], strongly indicating that there are autosomal recessive mutations in *EYS* yet to be identified. Keeping this in mind, when we specifically looked at retinitis pigmentosa patients who were still genetically unsolved after the NGS panel test[11], we

found that G843E was highly enriched in patients with a heterozygous deleterious mutation in *EYS* (allele frequency = 17.0%) compared to those without (allele frequency = 6.9%, odds ratio = 2.46, $P = 8.51 \times 10^{-7}$, Fisher exact test; Table 3) or to the general population using a public database (allele frequency = 1.7%, odds ratio = 10.0, $P = 2.21 \times 10^{-32}$, Fisher exact test; Table 3). This strongly suggests that the G843E allele contributes to retinitis pigmentosa in *trans* with another *EYS* mutation, as in ARRP. Similarly, the frequency of G843E homozygotes was significantly higher (odds ratio = 97.0, $P = 9.89 \times 10^{-12}$) in genetically unsolved retinitis pigmentosa patients (13/640) compared to the general population (1/4773) establishing that the G843E allele contributes to retinitis pigmentosa in homozygosity as well, typical for an ARRP mutation. Meanwhile, analysis of Peak 2, linked to S2556C, also revealed significant enrichment of the variant in unsolved patients with a heterozygous deleterious mutation in *EYS* compared to those without ($P = 2.56 \times 10^{-7}$, Fisher exact test), although the difference was relatively small (allele frequency = 39.1% vs 31.2%, odds ratio = 1.25). Taken together, the G843E mutation may cause retinitis pigmentosa when both alleles of *EYS* are affected, either in a compound heterozygous or a homozygous state, as observed in an ARRP allele. Meanwhile, Peak 2 may confer a different pathomechanism given the high frequency of the pathogenic allele (see "Discussion" section).

**Segregation analysis.** Although G843E is consistent with an ARRP variant according to the analysis above, it is unlikely that the G843E allele acts as a simple Mendelian allele, considering its relatively high allele frequency in the general population (1.7%). Theoretically, even homozygotes of G843E alone would cause ARRP in at least 1 in 3460 births, with a modest assumption of random mating, which is more frequent than the reported overall prevalence of ARRP in Japan (1 in 7000)[35]. Furthermore, although the allele frequency of G843E (1.7%) is 3.8-fold higher

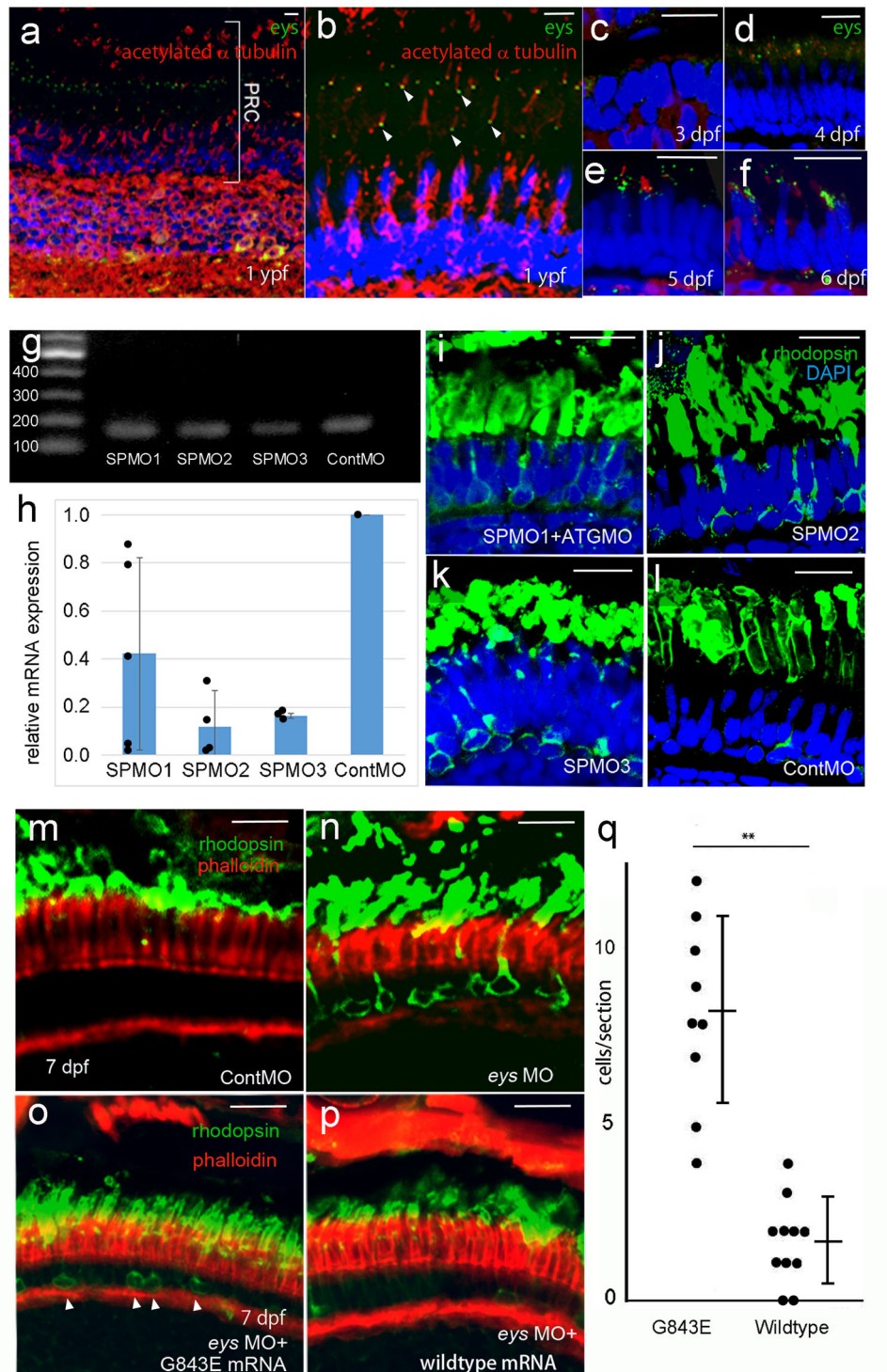

than that of the founder variant S1653Kfs (0.44%) in the general population (Table 2), the observed frequency of homozygotes of G843E (14/867) is actually lower compared to that of S1653Kfs (24/867) in retinitis pigmentosa patients. This could be accounted for by G843E causing incomplete penetrance or a mild retinal phenotype, both of which could lead to a large underestimation of disease frequency. To explore this possibility, we carried out a segregation analysis in 18 unaffected (and 1 affected) siblings of index patients with G843E (either in a compound heterozygous or a homozygous state, 13 families; Supplementary Fig. 2). None of the unaffected siblings of the patients carried biallelic

EYS mutations, except for the brother of YWC133, who was unexpectedly found to be compound heterozygous for G843E and S1653Kfs. This 75-year-old man was considered unaffected according to a local ophthalmologist who had carried out cataract surgeries on both eyes within the preceding year. Re-assessment of the patient at Tohoku University Hospital revealed a mildly but clearly constricted visual field, accompanied by moderate attenuation of the retinal vessels and diffuse alteration of the retinal pigment epithelium with modest retinal thinning in both eyes, although he had normal visual acuity (20/20). Nevertheless, the marked reduction in the electrical response of the patient's

**Fig. 4 Functional assessment of *EYS* G843E variant following morpholino-mediated knockdown of *eys* in zebrafish. a** Immunostaining of Eys (green) in zebrafish retina at 1-year post-fertilization (ypf). **b** High-magnification image of photoreceptors. Eys (arrowhead, green) localized at the basal side of connecting cilium (acetylated α tubulin, red) of the photoreceptors. **c–f** Expression of Eys during development at 3 days post- post-fertilization (pdf), 4, 5, and 6 dpf. **g** RT-PCR of eys at 4 dpf (45 cycles) following injection of three different MOs. **h** Quantitative RT-PCR analyses of the morphants (biologically independent samples). SP1MO ($N = 5$), SP2MO ($N = 3$), and SP3MO ($N = 3$). Eys expression was reduced by at least 50% at 4 dpf. Vertical bar: mean ± standard deviation. **i–l** Basal intracellular deposition of rhodopsin (rhodopsin mislocalization) observed following injection of three different MO at 6 dpf. Note, injections of three different MOs resulted in the same phenotype. **m, n** Rhodopsin localization in the photoreceptors at 7 dpf. **m** Rhodopsin is correctly localized at the photoreceptor outer segments in the control. **n** eys knockdown by MO-induced rhodopsin mislocalization toward the basal and the lateral membrane of the photoreceptors ($N = 6$ biologically independent fishes). **o, p** Greater improvement of the rhodopsin mislocalization was achieved in the eyes supplemented with wild-type human *EYS* mRNA (**p**) over those injected with mutant human *EYS* mRNA with G843E (**o**) after SPMO2-mediated knockdown of eys, consistent with decreased *EYS* function by the mutation. **q** A quantitative analysis of **o** ($N = 9$ biologically independent fishes) and **p** ($N = 9$ biologically independent fishes). Numbers of cells with mislocalized rhodopsin per retinal section were counted (vertical bar: mean ± standard deviation). The difference is significant ($P = 0.00903$; Wilcoxon rank-sum test) \*\*$P < 0.01$. PRC photoreceptors, Cont control. Scale bar = 10 μm (**a–e**, **i–p**).

**Table 3 Genotypes for G843E, S1653Kfs, and S2556C in *EYS*.**

| | Controls | Solved cases (N = 227) | | Unsolved cases (N = 640) | |
|---|---|---|---|---|---|
| | ToMMo | EYS | nonEYS | With EYS path mut | No EYS path mut |
| **Genotype count (G843E)** | | | | | |
| REF/REF | 4611 (96.6%) | 76 (96.2%) | 139 (94.6%) | 98 (66.7%) | 437 (88.6%) |
| REF/G843E | 161 (3.4%) | 3 (3.8%) | 8 (5.4%) | 48 (32.7%) | 44 (8.9%) |
| G843E/G843E | 1 (0.0%) | 0 (0.0%) | 0 (0.0%) | 1 (0.6%) | 12 (2.4%) |
| **Allele count (G843E)** | | | | | |
| REF | 9383 (98.3%) | 155 (98.1%) | 286 (97.3%) | 244 (83.0%) | 918 (93.1%) |
| G843E | 163 (1.7%) | 3 (1.9%) | 8 (2.7%) | 50 (17.0%) | 68 (6.9%) |
| MAF | 0.017 | 0.019 | 0.027 | 0.170 | 0.069 |
| **Genotype count (S1653Kfs)** | | | | | |
| REF/REF | 4,731 (99.1%) | 27 (34.2%) | 148 (100%) | 83 (56.5%) | 493 (100%) |
| REF/S1653Kfs | 42 (0.9%) | 29 (36.7%) | 0 (0.0%) | 64 (43.5%) | 0 (0.0%) |
| S1653Kfs/S1653Kfs | 0 (0.0%) | 23 (29.1%) | 0 (0.0%) | 0 (0.0%) | 0 (0.0%) |
| **Genotype count (S2556C)** | | | | | |
| REF/REF | 2949 (61.8%) | 62 (78.5%) | 84 (56.8%) | 50 (34.0%) | 152 (31.0%) |
| REF/S2556C | 1,605 (33.6%) | 13 (16.5%) | 58 (39.2%) | 79 (53.7%) | 261 (53.2%) |
| S2556C/S2556C | 219 (4.6%) | 4 (5.1%) | 6 (4.1%) | 18 (12.2%) | 78 (15.9%) |

ToMMo[51]: Normal Japanese population ($N = 4773$). EYS: retinitis pigmentosa patients genetically solved with biallelic *EYS* mutations. nonEYS: retinitis pigmentosa patients genetically solved with mutations in genes other than *EYS*. With EYS path mut: genetically unsolved retinitis pigmentosa patients with a heterozygous deleterious *EYS* mutation. No EYS path mut: genetically unsolved retinitis pigmentosa patients with no deleterious *EYS* mutation. The status of *EYS* mutations was extracted from a previous NGS panel test study[11].

retina to light stimuli probed by electroretinogram indicated that he also had a mild form of retinitis pigmentosa (Supplementary Fig. 3). Thus, the results are consistent with G843E being a hypomorphic ARRP allele and show that it can indeed sometimes cause mild retinal disease that may be overlooked without a thorough assessment. This may partly account for the gap between the known prevalence of retinitis pigmentosa and the allele frequency of G843E, complicating its interpretation in the past[11,13,36,37]. Assuming that G843E is an ARRP allele, the mutation would account for an additional 7.0% of Japanese cases of retinitis pigmentosa, which would increase the proportion of genetically solved cases by 26.8%, either as a compound heterozygotes or homozygotes.

## Discussion

Although previous reports have used GWASs to identify rare penetrant pathogenic variants in complex diseases[21–23], our study is the first to demonstrate that GWAS, with the help of the NGS panel test, can be applied effectively to identify genetic risks in heterogeneous monogenic disorders. We successfully identified three independent disease-associated signals, all in the gene *EYS*, including a signal in linkage disequilibrium with the known commonest founder mutation S1653Kfs in *EYS* that causes ARRP[11]. This confirmed the quality of GWAS and its ability to effectively detect classical Mendelian mutations, although S1653Kfs could have been identified with sequencing alone in this case. At the same time, the successful application of GWAS is dependent on there being common founders, which may limit its use in a highly heterogeneous population. Another important factor appears to be the study size as with the case of GWAS for common traits. While there are a few founder ARRP mutations including those in *EYS*, *USH2A*, *RP1*, *SAG*, and *RP1L1* reported in the Japanese population[4,11,15,38], those in *EYS* are by far the most frequent[11]. It is likely that this limited the GWAS to detect only exceedingly frequent founder mutations in *EYS*. However, increasing the number of cases and controls should greatly facilitate detection of less frequent founder mutations as demonstrated in recent large-scale GWAS studies that have boosted the number of disease-associated loci from a few initially to often dozens including those for glaucoma and age-related macular degeneration[21,39,40].

We detected a signal in linkage disequilibrium with G843E, a controversial variant that has been recognized by sequencing alone but did not previously fulfill the standard criteria required to determine pathogenicity[11,13,37,41]. Herein, we provide direct evidence of *EYS* dysfunction caused by G843E using zebrafish as a model. Furthermore, analysis of the NGS panel test data revealed that G843E was highly enriched in heterozygous carriers of another deleterious *EYS* mutation and homozygotes compared to the general population, consistent with the allele mediating the autosomal recessive mode of inheritance. Yet, the relatively high allele frequency of G843E contradicts the known prevalence of ARRP. A segregation analysis identified an elderly asymptomatic patient who was compound heterozygous for G843E and S1653Kfs and had been erroneously assigned as unaffected, probably based on a lack of symptoms or typical features of retinitis pigmentosa. This was consistent with G843E being hypomorphic sometimes causing a very mild phenotype later in life. In such instances, the disease may be overlooked without an assessment by electroretinogram, the most sensitive measure to

detect retinitis pigmentosa. Unfortunately, a reliable phenotypic comparison between G843E carriers versus non-carriers among patients with biallelic EYS mutations was not possible because many patients with G843E and milder phenotype are unlikely to be included in the genetic analysis, to begin with. This is supported by the presence of asymptomatic ARRP patient with G843E and S1653Kfs mutations (the brother of YWC133), who were erroneously assigned as unaffected prior to a thorough ocular examination and consistent with the disproportionally small number of G843E homozygotes relative to the expectation. Nevertheless, the strong evidence from the segregation analysis (P < 0.01), the presence of G843E in trans with an established pathogenic variant in multiple families, along with in vitro expression and in vivo functional analyses supporting dysfunction of G843E have allowed us to reclassify the variant as pathogenic according to the standard guidelines[41,42]. G843E as a quasi-Mendelian variant will likely enable genetic diagnosis in an additional 7.0% of Japanese patients with ARRP, which would represent a 26.8% improvement in the diagnosis rate. At the same time, the importance of this finding extended far beyond the context of genetic diagnosis as a detection of a founder mutation with an extremely large disease contribution provides a unique opportunity for the development of an AAV-mediated mutation replacement genome-editing gene therapy, which has shown promising in vivo outcomes[15,16,43]. This demonstrates the robustness of the approach, considering that mutations in novel retinitis pigmentosa genes, which are still continuously discovered by sequencing, rarely account for more than 1% of cases and are unlikely to be suitable targets for drug development because of the extremely low number of the patients affected.

Recently, enrichment of the G843E variant in EYS in a group of patients with hereditary retinal degenerations (HRD) that carried a quasi-Mendelian allele in another gene (c.5797 C>T/p.R1933* in RP1), has been reported[4], suggesting indeed a non-Mendelian, oligogenic or genetic modifier role of EYS in retinal degeneration. In our study, RP1-R1933* was infrequent among carriers of EYS-G843E, but this may be attributable to gross differences in the clinical phenotypes considered (macular degeneration or cone-rod dystrophy in the previous reports[4,44] vs. canonical ARRP, studied here) Furthermore, while in heterozygous carriers RP1-R1933* seems to exert its pathogenic functions via the co-presence of EYS-G843E and other hypomorphic alleles outside of the RP1 locus[4], a reciprocal mechanism is not forcibly true, since molecular pathology of EYS-G843E in ARRP may follow different routes, as clearly shown above. Taken together, these results emphasize the unexpected pleiotropic role of EYS-G843E with respect to the range of unconventional genetic influence and its effect on clinical phenotypes.

GWAS also identified a novel retinitis pigmentosa-associated EYS signal (Peak 2) with no rare exonic or splice site variants in linkage disequilibrium that could account for ARRP. It is possible that another quasi-Mendelian mutation in linkage disequilibrium with Peak 2 in the non-coding regions remains undetected after NGS panel test[4,7,19]. However, the higher frequency of the lead variant (allele frequency = 0.216) for this peak is distinct from those of the other peaks (allele frequency = 0.041 and 0.0005), resulting in a lower OR (1.83) well within the range of those for more common retinal diseases such as age-related macular degeneration[45]. Therefore, it is possible that the true pathogenic variant(s) in linkage disequilibrium may be a high-frequency variant(s) behaving in a non-Mendelian manner, similar to those presumed to account for susceptibility signals in common diseases although this is less likely given that retinitis pigmentosa is a rare disease. For example, the risk variant may act in an oligo-genic fashion or as a disease modifier in combination with mutations in other genes. At the same time, it is possible that the

unknown true pathogenic variant(s) different from the S2556C variant lies deep in a non-coding region as typical for signals detected by GWAS in common diseases. Although the exact mode of genetic influence remains to be elucidated for this peak, the findings stress the importance of breaking the stereotypical dogma of Mendelian inheritance in monogenic diseases and emphasize the importance of large-scale genome-wide case–control genetic studies in elucidating the genetic causes of inherited diseases largely unsolved by sequencing approaches.

In conclusion, this study demonstrates the usefulness of GWAS in identifying disease-associated loci, in so-called monogenic disorders, which is dependent on the presence of founder mutations. It also highlights the under-appreciated significance of high-frequency variants that may account for the undetermined heritability of various inherited diseases. At the same time, the significance of the identified variants may extend beyond genetic diagnosis as they may simultaneously serve as ideal targets of local genome treatments.

## Methods

**Patients and controls**. Nine-hundred forty-four presumed-unrelated patients with retinitis pigmentosa were recruited from Kyushu University Hospital, Tohoku University Hospital Yuko Wada Eye Clinic, Nagoya University Hospital, and Juntendo University Hospital. The majority of the patients were recruited through a genetic screening project hosted by the Japan Retinitis Pigmentosa Registry Project (JRPRP) in which 83 genes associated with retinitis pigmentosa were analyzed by NGS panel test (sequencing of all the coding exons and the flanking immediate exon–intron boundaries of the known 83 retinitis pigmentosa genes)[11]. Japanese patients consistent with the autosomal recessive mode of inheritance that typically had multiple affected siblings and no affected members in other generations were enrolled. In addition, isolated cases with no family history as well as the offspring of consanguineous parents were also included. Most of the unaffected controls, who were ruled out for retinitis pigmentosa with a fundus examination, were recruited at Tohoku University Hospital and its affiliated hospitals[39]. The remaining control samples from subjects with no documented history of ocular disease were purchased from the National Institutes of Biomedical Innovation, Health and Nutrition (https://bioresource.nibiohn.go.jp/). Blood samples were collected for DNA extraction and establishing patient-derived lymphoblastoid cell lines (LCLs).

**Genome-wide association study**. In the first GWAS, 644 cases and 620 controls, all from Japan, were genotyped with the CoreExome-24 v1.1 (Illumina, San Diego, CA, USA). The total number of analyzed samples was reduced to 581 cases and 603 controls after quality control (QC). During QC, we excluded single-nucleotide variants (SNVs) with Hardy–Weinberg equilibrium (HWE) P < 0.0001 in the controls, a call rate <99%, or three alleles. Data were also discarded if the sample had a call rate <98%. In addition, closely related pairs (pi-hat > 0.1)[46], or ancestral outliers, as determined with a PCA analysis using the 1000 Genomes Project (five Asians, CEU, and YRI) and PLINK software were removed. One hundred and forty-nine cases with causal mutations identified after the NGS panel test[11] were also removed. The remaining 432 cases and 603 controls were subjected to a GWAS using 10,673,864 variants following whole-genome imputation of 523,187 genotyped SNVs using phased haplotypes from the 1000 Genomes Project (Phase 3) as the reference panel. SHAPEIT was used for phasing, followed by minimac3 for genotype imputations[47]. Imputed variants with estimated imputation accuracy of Rsq >0.3 were selected. It should be noted that the variants were not excluded based on minor allele frequency (MAF) in this study because assumed rare retinitis pigmentosa mutations may be tagged better with lower frequency variants. Statistical analysis of the GWAS was performed using RVtests[48]. We used imputed genotype dosages and top 10 principle components as covariates for the analysis input data. The principal component scores were calculated using PLINK. The association between each SNP and retinitis pigmentosa was modeled as logistic regression with an allele dosage effect and adjusted for the 10 principle component scores. The Wald test was used to determine the significance of association for each SNP.

The second GWAS comprised 300 cases and 300 controls and was also carried out using genotyping with the CoreExome-24 v1.2 (Illumina). The total number of samples was reduced to 286 cases and 287 controls after applying QC procedures identical to the first GWAS. Samples were also removed if they overlapped with the first GWAS. Then, 78 cases with causal mutations identified through the NGS panel test were excluded[11]. The remaining 208 cases and 287 controls were subjected to a GWAS using 10,383,808 SNVs following whole-genome imputation of 522,207 genotyped SNVs selected with the same criteria as the first GWAS. Since it was very difficult to estimate the outcome initially because we could not find a GWAS targeting recessive Mendelian disorders, we estimated the size of the second GWAS based on the results of the first GWAS assuming that meta-GWAS was to

be performed. The first GWAS was carried out using all the samples available at that time. Sample sizes for the second GWAS were calculated so that the top five signals would reach statistical significance using an online sample size calculator (https://www.stat.ubc.ca/) adopting a two-sided alpha-level of 0.05, 80% power. However, the size was eventually restricted by the availability of the samples because the disease studied was a rare disease with a prevalence of 1 in 4000. Of the cases and controls actually used in GWAS, 284, 284, and 72 cases used were recruited via Tohoku University (northeastern region), Kyushu University (southern region), and Nagoya University (central region) whereas 797 and 93 controls were from Tohoku University and unknown region (purchased as a normal Japanese DNA set), respectively.

A meta-analysis combining the first and second GWAS data sets was performed using METAL[49]. Stepwise conditional analysis has been used as a tool to identify secondary association signals at a locus[50]. The conditional analyses, starting with a top associated variant, were performed using the dosages of target variants of the regions used as covariates. These steps of adding the variant dosages to the covariate one by one were repeated until there were no variants satisfying the genome-wide significance level ($P < 5.0 \times 10^{-8}$).

To assess the linkage between non-synonymous variants identified through a previous NGS panel test[11] and the GWAS peak variants positioned within 1.5 Mb of each other, correlation coefficients ($r^2$) and D'/LOD were calculated using Haploview. Allele frequencies of variants in the Japanese general population were estimated using ToMMo (https://www.megabank.tohoku.ac.jp/english/), a genomic database from whole-genome sequencing of 4773 Japanese healthy individuals which has recently expanded from 3552[51].

**WGS and Sanger sequencing**. We performed WGS using the NovaSeq 6000 (Illumina, San Diego, CA, USA) sequencer with 151 bp paired-end reads. The sequencing library was constructed using the TruSeq Nano DNA Library Prep Kit (Illumina) according to the manufacturer's instructions. The sequenced reads were aligned to the human reference genome using BWA-mem (ver. 0.7.17). Then, PCR duplicate reads were marked using Picard tools (ver. 2.17.8). Base quality scores were recalibrated, and SNVs and short insertions and deletions were called, using GATK (ver. 4.1.2.0) according to the GATK Best Practices (https://software.broadinstitute.org/gatk/best-practices/). Structural variants were called using Manta[52] according to the instructions commands for the Single Diploid Sample Analysis (https://github.com/Illumina/manta/blob/master/docs/userGuide/README.md). In addition, we used IGV software to visually inspect reads for specific genes that were reported to carry structural variants.

Sanger sequencing was carried out for genotyping of family members using the protocol described earlier[53]. In brief, genomic DNA was amplified with PCR using Amplitaq Gold and a primer pair designed by Primer3 (ver. 0.4.0; http://bioinfo.ut.ee/primer3-0.4.0/). PCR amplification was performed in a 20 μl total volume containing 20 ng genomic DNA, 1× GoTaq buffer, 0.5 mM dNTPs, 10 μM of each primer, and 2 units (5 U/μl) of GoTaq polymerase (Promega, Madison, Wisconsin). The PCR amplicons were applied onto a 2% agarose gel with appropriate controls and markers.

**mRNA analysis using patient-derived lymphoblastoid cell lines**. To generate patient-derived LCLs, lymphocytes were transformed with the Epstein-Barr virus at a core facility run by Tokyo Medical Dental University. The origin of the LCL cells was identified as patients by partial sequencing of the genome. LCLs were cultured in RPMI1640 medium (ThermoFisher Scientific, Waltham, MA) supplemented with 15% fetal bovine serum (FBS; ThermoFisher Scientific), 2mM L-glutamine (ThermoFisher Scientific), and 1% penicillin/streptomycin (ThermoFisher Scientific) at 37 °C in an atmosphere of 5% $CO_2$. Cells were tested routinely for mycoplasma contamination. A plasmid for CAG promoter insertion genome editing[50] was constructed as shown in Supplementary Fig. 4A, B. The donor template, which comprised the flanking micro-homology arms, gRNA target site, and the donor sequence, were sub-clone and inserted into the single CRISPR/Cas9 vector (pX601, addgene #61591). gRNAs were designed (Supplementary Fig. 4C) and T7E1 assay (Supplementary Fig. 4D) were performed as the manufacturer's instructions (New England Biolabs, Ipswich, MA). In brief, PCR products amplified using genomic DNA were denatured at 95 °C for 5 min, reannealed, and incubated with T7 Endonuclease I (New England Biolabs) at 37 °C for 30 min. The reaction products were resolved by electrophoresis in 2% agarose gel. DNA fragments were analyzed using ImageJ. The indel efficiency was calculated as $100 \times (1 - (1 - \text{cleaved band intensity/total band intensities})1/2)$. The donor sequence included a CMV promoter (from pCAG-Neo, Wako, Osaka, Japan) for in-frame insertion upstream of the EYS start codon. A plasmid for mutation replacement genome editing was constructed as shown in Supplementary Fig. 4E, F. The donor template, which comprised the flanking micro-homology arms, gRNA-1 target site or gRNA-4 target site, and the donor sequence, were sub-cloned and inserted into the vector (pX601) using a DNA ligation kit (Clontech, Mountain View, CA). To avoid template cleavage after mutation replacement, mutations were introduced in the flanking gRNA target sites within the donor template. The mutations introduced in the 5′ gRNA-1 and 3′ gRNA-4 target sites were selected using codon optimization tool GENEisu (http://www.geneius.de/GENEius/) on human codon table. The LCLs were transfected with a plasmid using Trans-IT XP transfection reagent (Mirus Bio, Madison, WI) treated with or without a demethylating agent, 5-Aza-2′-deoxycytidine (1 μM; Abcam,

Cambridge, UK), and hydralazine hydrochloride (0.2 μM; Abcam). To test whether transcripts were degraded by nonsense-mediated mRNA decay, LCL was treated by emetine (Sigma-Aldrich, St. Louis, MO) at 60 μg/ml for 12 h before RNA extraction[54]. For mutation replacement gene editing (Supplementary Fig. 3E, F), LCL was co-transfected with the CAG promoter insertion plasmid and the mutation replacement genome-editing plasmid (ratio 1:3). Total RNA was extracted 48 h post-transfection using the miRNeasy plus mini kit (Qiagen, Hilden, Germany) according to the manufacturer's instructions. A 500-ng sample of total RNA was reverse-transcribed with SuperScript IV (ThermoFisher Scientific) and oligo(dT) primers (ThermoFisher Scientific) at 55 °C for 30 min. The design of the primer sets for RT-PCR is shown in Supplementary Table 5. The RT-PCR reaction was performed with KOD One DNA polymerase (Toyobo, Osaka, Japan) at 35 cycles of 98 °C for 10 s, 60 °C for 5 s, and 68 °C for 5 s. PCR products were analyzed on agarose gels. Uncropped gel images were provided as Supplementary Fig. 5.

**Zebrafish experiments**. Zebrafish (*Danio rerio*) AB strain was maintained and bred in a standard fashion[55] with 14 h light/10 h dark cycle. Morpholino oligonucleotides and eys mRNA microinjection were performed on 1–8 cell stage embryos using micro-needle injection[56,57]. Morpholino oligonucleotides were obtained from Gene Tools LLC. The following morpholinos were used: ATGMO, 5′-CTCATGTTTGTCTTGGCTCGACTGG-3′; SPMO1, 5′-TTGACTTACCCTTA AATCCTGGTG-3′; SPMO2, 5′-AAAGTTCCTTCACTGTGAATGGAGC-3′; SPMO3, 5′-CAGAGCAGTTGACACCTATGATGTG-3′, standard control morpholino, 5′-CCTCTTACCTCAGTTACAATTTATA-3′. RT-PCR and Quantitative RT-PCR analyses were performed using the following primers. Eys201F 5′-ACGTTGA CGGATGCTATGAGCAG-3′ Eys201R 5′-AGCAAGCTCCTTCTTTGCACC-3′. DrGAPDHF1 5′-TCACACCAAGTGTCAGGACG-3′, DrGAPDHR1 5′-CG CCTTCTGCCTTAACCTCA-3′.

Total RNA was isolated from each morpholino injectants (at least 7 embryos) according to the manufacturer's instructions, which was resuspended in 20 ml of RNase-free water. After the RNA was treated with DNase, we further extracted the RNA with phenol–chloroform. Then, RNA was reverse-transcribed with the SuperScript VILO cDNA Synthesis kit (Life Technologies®, ThermoFisher Scientific). The RT products (cDNA) were amplified and analyzed by quantitative real-time PCR (Applied Biosystems 7500 fast real-time PCR System, Applied Biosystems®, ThermoFisher Scientific) using TAKARA SYBR Green PCR mixture® (TakaRa Bio Inc., Kusatsu, Shiga, Japan). The expression of the *eys* was normalized by the comparative threshold cycle method and internal control (*gapdh*) and calculated by the ΔΔCt method.

For Human EYS cDNAs with and without c.2528 G>A variant were sub-cloned into pcDNA 3.1(+) vector and were transcribed by the mMessage Machine T7 kit (Ambion®, ThermoFisher Scientific, Waltham, Massachusetts, USA). In morpholino knockdown experiments with or without rescue mRNA, a mixture of 200 mM ATG-MO, SP-MO, and 300 ng/μl mRNA or 380 mM ATG-MO was applied, respectively. We injected morphorinos and mRNA into embryos within 40 min after fertilization.

After killing the fish, zebrafish embryos or adult head were placed in 4% (w/v) paraformaldehyde (PFA), pH 7.4, in PBS overnight at 4 °C, and then incubated in 20% glucose aqueous solution overnight at 4 °C. The fixed fish were embedded in an optimal cutting temperature compound (Neg50TM, ThermoFisher Scientific) and quick-frozen in a −150 °C nitrogen freezer. The samples were cut into 14–16 μm sections. The sections were placed in a solution containing 0.1 M PBS, 10 % BSA for 2 h at room temperature, and then incubated with the primary antibody at 4 °C overnight. After a 0.1 M PBS/0.005% Tween rinse cycle, the sections were incubated with Alexa Fluor 594-conjugated IgG antibodies (1:500, Jackson ImmunoResearch, West Grove, Pennsylvania, USA) species-matched for the primary antibody, rhodamine-conjugated phalloidin (1:200, Cytoskeleton Denver, Colorado, U.S.A.), or DAPI (1:1000, Cytoskeleton) at room temperature for 2 h. The slides were then mounted with an aqueous mounting medium (PermaFluor®, ThermoFisher Scientific) after another cycle of rinsing. The sections were analyzed with the use of the confocal fluorescence laser microscopes LSM 710 (Carl Zeiss, Jena, Thüringen, Germany). The following primary antibodies and dilutions were used: rabbit polyclonal anti-Eys antibody (1:200, Novus Biologicals, Centennial, Colorado, USA), rabbit polyclonal anti-rhodopsin (bovine) antibody (1:500, Abcam, Cambridge, United Kingdom), and mouse monoclonal [6-11B-1] anti-alpha Tubulin (acetyl K40) antibody (1:200, Abcam, Cambridge United Kingdom).

The Quantitative analyses of mislocalized rhodopsin were assessed by the confocal microscope. The total number of photoreceptors that displayed rhodopsin mislocalization within a retinal section was counted by a person who was masked with regard to what was injected.

**Statistics and reproducibility**. The frequency of homozygotes of the G843E and S1653fs mutations was calculated as the square of allele frequency of each mutation in the general population, assuming random mating. Numbers of cells with mislocalized rhodopsin in the field were counted and the Wilcoxon rank-sum test was carried out using R software to assess the difference between the two groups. RT-PCR was confirmed to be reproducible by three independent assays. In zebrafish experiments, three blinded observers independently analyzed the data.

**Ethics approval and consent to participate**. The study was initiated after ethical approvals were granted by the Institutional Review Boards of Kyushu University Hospital, Tohoku University Hospital, the Yuko Wada Eye Clinic, Tokyo Medical and Dental University, and Nagoya University Hospital. All procedures followed the tenets of the Declaration of Helsinki. Informed consent was obtained from all patients and controls before collecting blood samples for DNA extraction and establishing patient-derived LCLs. All zebrafish experimental procedures were conducted after approval by the related committees, including the animal ethics committee, for the animal experiments at Osaka University Graduate School of Medicine.

**Reporting summary**. Further information on research design is available in the Nature Research Reporting Summary linked to this article.

## Data availability

GWAS summary statistics have been uploaded to the GWAS catalog (https://www.ebi.ac.uk/gwas). Study Accession IDs are GCST90011892, GCST900118923, and GCST90011894 for 1st set, 2nd set, and metadata. Raw data used during and/or analyzed for all Figures and Tables are available from the authors within the limits of the Institutional ethical and research approvals, upon a reasonable request. Raw data for Fig. 4h, q are available in Supplementary Data 1.

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

## Acknowledgements

The study was supported by the Japan Retinitis Pigmentosa Registry Project. We thank J. Inazawa and M. Takaoka (Bioresource Laboratory, Tokyo Medical and Dental University) for technical support in establishing the patient-derived LCLs. This study was supported in part by the Japan Agency for Medical Research and Development (#19ek0109213h0003, K.M.N. and #JP18lk1403004, T.A.) and Swiss National Science Foundation (#31003A_176097, C.R.).

## Author contributions

K.M.N. designed the overall study. Collection of DNA samples and clinical assessment was performed by K.M.N., H.K., T.A., S.U. and Y.I., and was led by Y.W., A.M., H.T., K.-H.S., and T.N. F.M., Y.K., K.C., J.N., M.A. and T.K. performed the bioinformatics analyses. K.F. and M.T. performed LCL studies. Zebrafish experiments were carried out by Y.M., T.T. and M.T. Sequencing was carried out by F.M., K.F., K.S., M.T. and Y.M. K. M.N., C.R., T.T. and T.N. provided the funding. All authors read and approved the final manuscript.

## Competing interests

K.M.N., K.F., and T.N. are listed as inventors in a patent application for genome-editing gene therapy (PCT/JP2019/ 43905). The remaining authors declare no competing interests.

## Additional information

[1]Department of Ophthalmology, Tohoku University Graduate School of Medicine, Aoba-ku, Sendai, Japan. [2]Department of Advanced Ophthalmic Medicine, Tohoku University Graduate School of Medicine, Aoba-ku, Sendai, Japan. [3]Department of Ophthalmology, Nagoya University Graduate School of Medicine, Showa-ku, Nagoya, Japan. [4]Department of Medical Science Mathematics, Medical Research Institute, Tokyo Medical and Dental University, Bunkyo-ku, Tokyo, Japan. [5]Laboratory for Medical Science Mathematics, RIKEN Center for Integrative Medical Sciences, Tsurumi-ku, Yokohama, Japan. [6]Laboratory for Medical Science Mathematics, Department of Biological Sciences, Graduate School of Science, The University of Tokyo, Bunkyo-ku, Tokyo, Japan. [7]Department of Biomedical Informatics, Osaka University Graduate School of Medicine, Suita, Japan. [8]Department of Ophthalmic Imaging and Information Analytics, Tohoku University Graduate School of Medicine, Aoba-ku, Sendai, Japan. [9]Laboratory for Statistical Analysis, Center for Integrative Medical Sciences, RIKEN, Tsurumi-ku, Yokohama, Japan. [10]Department of Ophthalmology, Graduate School of Medical Sciences, Kyushu University, Higashi-ku, Fukuoka, Japan. [11]Division of Pulmonary Medicine, Department of Medicine, Keio University School of Medicine, Shinjuku-ku, Tokyo, Japan. [12]Collaborative Program for Ophthalmic Drug Discovery, Tohoku University Graduate School of Medicine, Aoba-ku, Sendai, Japan. [13]Unit of Medical Genetics, Department of Computational Biology, University of Lausanne, Lausanne, Switzerland. [14]Department of Ophthalmology, Juntendo University Graduate School of Medicine, Bunkyo-ku, Tokyo, Japan. [15]Division of Clinical Cell Therapy, United Centers for Advanced Research and Translational Medicine (ART), Tohoku University Graduate School of Medicine, Aoba-ku, Sendai, Japan. [16]Laboratory for Genotyping Development, RIKEN Center for Integrative Medical Sciences,, Tsurumi-ku, Yokohama, Japan. [17]Yuko Wada Eye Clinic, Aoba-ku, Sendai, Japan. [18]Clinical Research Center, Institute of Molecular and Clinical Ophthalmology Basel (IOB), Basel, Switzerland. [19]Department of Ophthalmology, University Hospital Basel, Universitätsspital CH, Basel, Switzerland. [20]Department of Genetics and Genome Biology, University of Leicester, Leicester, UK. [21]Department of Ophthalmology, Miyazaki University School of Medicine, Miyazaki, Japan. [22]These authors contributed equally: Koji M. Nishiguchi, Fuyuki Miya. ✉email: kmn@med.nagoya-u.ac.jp

