## [Peer Review File · Communications Biology]

Reviewers' comments:

Reviewer #1 (Remarks to the Author):

In this paper the authors use GWAS to identify three association peaks within the EYS gene and study their causality and population impact. The approach is innovative, but might not be generally applicable. The results are believable, the analysis is generally cogent, and the Discussion is informative and complete. The most important contributions of this work are the functional studies of the B843E mutant in zebrafish, and the application of a population genetic approach and segregation analysis to show that it is actually probably causative, both of which are very interesting. Specific comments follow:

1. Abstract, lines 63-66, "Two were each tagged by a low frequency variant (allele frequency < 0.05); a known founder Mendelian mutation (c.4957dupA, p.S1653Kfs*2) and a presumably hypomorphic non-synonymous variant (c.2528G>A, p.G843E)." This sentence is a little unclear--are c.4957dupA and c.2528G>A the two tagged variant? If so, moving them to immediately behind Two might clarify it ('Two, a known founder...').
2. Introduction, lines 93-95, "Furthermore, there is a growing interest in the detection of prevalent founder mutations, which are also potential targets of AAV-mediated genome-editing therapy.": If there is a particular reason founder mutations would be good targets for AAV mediated therapy, the authors should probably state it, because it is not clear to the average reader.
3. Results, Detection..., p. 7, line 118, "We gathered a total of 944 DNA samples from unrelated patients...": The authors should include a bit more description of the patient group regarding ascertainment centers and inclusion criteria and comparison of the demographics of the two groups and controls.
4. Results, p. 8, lines 128-129, "The results of the two GWASs are summarized in Figure S1 and Tables S2 and S3.": The authors need to provide some additional detail on the two initial GWAS screens including a summary of the actual genotypes, the test performed including the model tested (e.g., recessive, genomic, a simple allelic test--most likely, etc.), and the type of correction carried out for multiple testing. From just looking at the data from the first two GWAS, it would appear that they would be suggestive rather than significant if a Bonferroni correction were applied. Is this the case? Or are the authors correcting only for the SNPs actually genotyped, since the remainder were imputed and thus were not independent estimates?
5. Results, p. 8, lines 134-136, "Subsequent conditional analysis of the EYS locus detected 3 independent genome-wide significant signals ($P < 5.0 \times 10^{-8}$; Peaks 1 - 3 in the order of significance, Table 1 and Figure 1B).": The authors could expand the explanation of how this was done--what constituted the conditional analysis and how it differed (and improved) the initial analysis.
6. Results, p. 9, lines 149-150, "...registered as benign/likely benign in the ClinVar (<https://www.ncbi.nlm.nih.gov/clinvar/>).": How was this change evaluated by the other commonly used estimators (Polyphen2, Mutation Taster, SIFT, etc.)? Also, what is the frequency of homozygotes in the Japanese population? If it is in HWE and has a high penetrance as expected for Mendelian diseases, homozygotes for this change would give too many cases by an order of magnitude. If it is hypomorphic, is there any difference in the phenotype of patients with this variant (e.g., age of onset, rate of progression, etc.)?
7. Results, p. 8, lines 140-144, "Peak 1 (rs76960013, AF = 0.0414, odds ratio (OR) = 3.95, $P = 1.18 \times 10^{-13}$) was in linkage disequilibrium (LD; $r^2 = 0.68$) with c.2528G>A (p.G843E; hereafter termed G843E; Table 1). G843E with an AF (0.0171) unusually high for ARRP has been described in conflicting ways in past reports, as having uncertain significance 23, being non-pathogenic 24, possibly being pathogenic...": The authors are correct that this also seems very high, although they deal with this well later on. Is this the case in published screens? Similarly, are the cases associated with this variant distinctive relative to the rest, especially regarding severity of the RP? Can you expand the analysis of family YWC133 to additional patients as this would be highly convincing. Are there any clinical data on the 12 G834E homozygotes with arRP in Table 2? One might expect them to be milder, with later onset and slower progression, etc.
8. Results, p. 9, lines 152-153, "It was detected even after removing solved cases with biallelic EYS mutations...": What do the authors mean by 'detected'? rs79476654 by itself only gave a $p = 2.54 \times 10^{-8}$, which is barely significant after correction.
9. Results, p. 9, general and Table 1: Were the 3 SNP association peaks in linkage disequilibrium

with each other? Fig. 1C is helpful, but an explicit test would be nice to see.

10. Table 1: It would help to see the p values for the linked variants directly. They should, in theory, be as high or higher than the corresponding associated SNPs.

11. Results, Detection..., general: It would be helpful for the authors to expand the description of the association studies of each of the putatively associated SNPs to a haplotype of high LD SNPs, as this would help to argue against spurious association. Association of the causative variation with a SNP also implies a founder effect, which could be verified by identifying an associated haplotype. Also, if the 3 associated SNPs are not in LD, or if they are part of separate risk haplotypes, as appears from Fig. 1C, it would help argue for identification of 3 independent and separate loci, even if they are within the same gene.

12. Results, Expression..., p. 11, lines 189-192, "These results favor against the presence of an intronic mutation in LD with Peak 1 that results in altered splicing and a presumed premature termination of the reading frame, but support G843E as the causal mutation linked to Peak 1.": These results do argue against this change causing a change in splicing, but do not address a change in expression, or decay due to an associated extraneous sequence change. Did the authors consider the possibility of the G843E change causing decreased expression through alteration of codon-usage?

13. Results, Expression..., p. 11, lines 181-192, "...transcription of EYS...Peak1.": This paragraph, and its relation to Fig.s 2 and S3, are a little unclear. First, please define LCL at its first use. Do untreated LCLs not express EYS mRNA as it would appear from Fig. 2B? Demethylation appears sufficient for expression of exons 4-5 and 14-18, but not 40-43 in wt and G843E cells, is this correct? If so, why? If the S1653fs mutation undergoes NMD, as suggested by emetine treatment, why are exons 4-5 and 14-18f detected without treatment? Is there a small amount of mRNA that escapes decay or undergoes an alternate splicing pathway, or is Tv6-3 or 7 perhaps expressed in the lens? Finally, Fig. S3C-F, explaining replacement of the S1653fs mutation, do not appear to be referenced in the text.

14. Results, Functional..., p. 12, lines 201-202, "During development, Eys expression was observed after 4 days post-fertilization (dpf, Figure 3D-F).": It looks like there might be some expression at 3 dpf. Is this the case? Did the authors do a quantitative test?

15. Results, Functional..., p. 11-12, general: While the results obtained with morpholino knockdown and rescue are convincing to this reviewer, they do not meet the current guidelines for morpholino use in zebrafish (Didier et al., PLoS Genetics, 2017), which would require knockout confirmation. The authors should address this either by doing the knockout and rescue or justifying not doing so in the Discussion.

16. Results, Enrichment..., p. 13, lines 219-220, "This establishes that the G843E allele contributes to RP in trans with another EYS mutation, as in ARRP.": The authors might consider replacing 'establishes' with 'strongly suggests' or something similar.

17. Results, Enrichment..., p. 13, line 230, "Meanwhile, Peak 2 may to confer a distinct pathomechanism.": This needs to be explained and expanded.

18. Results, Enrichment, p. 14, lines 247-255, "None of the unaffected...mild form of RP": It would really strengthen the manuscript to show the detailed clinical data from this patient, and any others with homozygous or heterozygous G843E mutations.

19. Discussion, p. 16, lines 270-271, "This confirmed the quality of GWAS and its ability to effectively detect classical Mendelian mutations.": This is a little unclear from the results presented. While the GWAS was useful in that it pointed to 3 variants, each of these variants would have been identified (or perhaps were identified previously) by WES or WGS. The authors might wish to soften this statement. They might also mention that the use of GWAS is dependent on there being a common founder for each mutation, so that this will not work in all cases, especially if there are multiple independent mutations causing the RP.

20. Discussion, p. 19, lines 325-329, "In conclusion, this study provides a novel GWAS-based framework for systematically detecting disease-associated variants, unbiased with regard to genomic location and mode of genetic influence, in so-called "monogenic" disorders. It also highlights the under-appreciated significance of non-Mendelian high frequency variants that may significantly account for the undetermined heritability of various inherited diseases.": It is actually unclear from the results presented here that the GWAS was necessary for the most important findings presented in this paper, which are the roles of the p.S1653Kfs*2 and especially the p.G843E mutation in their series of patients. These were detected in their sequencing work, perhaps before the GWAS was performed, and had been reported as founder mutation and suspect mutation before. In addition, one of these variants is a classical high-penetrance Mendelian allele,

while the second is still Mendelian and monogenic, but simply has reduced penetrance. These conclusions need to be revised a bit, perhaps acknowledging the usefulness of the association results in identifying 3 discrete regions within EYS and also mentioning the importance of a founder effect for application of association studies using non-causative markers.

Suggestions for English grammar and usage:

1. 84 "presenting with a various hereditary pattern": maybe 'variety of hereditary patterns'
2. 99 "Genome-wide association study (GWAS) is a type of analysis": maybe '...studies are a...'
3. 189 "These results favor against...": Maybe 'argue against' or 'favor causality of the G843E change over...' or something similar.
4. 195 Among mammals, only primates have EYS gene.: Maybe 'have the EYS gene' or 'have EYS'.
5. 201 "...photoreceptors in adult fishes (Figure 3A, B).": Maybe '...adult fish...'. 'Fishes' is usually used to refer to multiple species of fish, while the plural of fish is usually fish.
6. 302 "among carriers of EYS-G843E, but this maybe attributable": Maybe 'may be' instead of 'maybe'.
7. 321 "finding stress the importance": Maybe 'finding 'findings stress' and change '...and emphasize...'

Reviewer #2 (Remarks to the Author):

This is a unique, well written, and interesting paper in which the authors use GWAS in a cohort of patients with RP in which the genetic etiology of disease has not been found. GWAS revealed variants in EYS, at least 2 of which helped explain disease in a subset of the patients.

What remains unclear is what sequencing was performed in these patients prior to GWAS (the term "targeted re-sequencing" is mentioned several times throughout the manuscript, but I can't see exactly what this is. What genes were sequenced and why was it "re-sequencing?" Were they previously sequenced before the re-sequencing? Was NGS panel testing performed at any point?), and why was EYS the only gene that was found on GWAS out of the 80 plus genes that are known to cause RP?

The main takeaway from this paper is that the G843E variant, previously of unknown significance, now has good evidence for pathogenicity, including functional data in zebrafish and segregation data, in addition to the GWAS evidence. The pathogenicity of S2556C remains unclear, and S1653Kfs was already known to be pathogenic. It is interesting that GWAS can be used to help provide evidence for pathogenicity of variants, but the authors need to address why they think EYS was the only gene found, and whether that raises any concerns. For example, do they think EYS is by far the most common cause of ARRP in unsolved cases? Relevant to this question is what genetic testing they already had (again, what was the "targeted re-sequencing" they had?).

In addition, several point by point comments are below. I have given fairly extensive comments and revisions, but overall I recommend publication of this unique paper.

1. Abstract line 60, uncover "the" genetic basis
2. Abstract line 61, applied "a" 2-step genome-wide association
3. Abstract line 65, I would say "possibly" hypomorphic instead of "presumably" hypomorphic. Later you present evidence for why you think it is hypomorphic, but here in the abstract it is unclear why you are presuming it to be hypomorphic.
4. In the abstract, line 68, I would say "possibly" not likely, as will be discussed in the points below. Given high AF and lack of functional data or segregation data like you have for the G843E, I think the role of this variant in disease is still very unclear.
5. Introduction line 79, what is meant by "simplex sequence-based approach?" Please reword or remove. Do you mean the "traditional next-generation sequencing approach?"
6. What is "targeted re-sequencing." Is this sanger sequencing of specific genes? Is it just exons? Exons plus exon-intron boundaries? The entire gene? And why "re"? Was this gene sequenced in these patients previously?
7. When you refer to allele frequencies, what database are you using?
8. Abstract line 68, you say the third peak is tagged by an intronic common variant, but later when you say the 3 variants, none of them seem to be intronic. 2 are missense mutations (G843E and S2556C) and 1 is a frameshift (S1653Kfs). Please clarify.

9. Somewhere in the results, I would like to see, for each of the 3 variants, how many of the total RP cases in the cohort had that variant, and of those, how many were homozygous, how many were heterozygous, and of the heterozygotes how many had a second known pathogenic or likely pathogenic variant in EYS? In other words, if there were 432+208 cases analyzed (640), say 50 of them had the first variant (G843E), and 20 of those were homozygous, and the other 30 were heterozygous. Of the 30 heterozygotes, say 10 had a second known pathogenic or likely pathogenic variant in EYS. That is useful information, because now I know that 30 of those 640 RP cases are potentially explained by EYS mutations. Without those numbers, it makes it harder to make sense of the results.

10. In the Discussion, you should point out that these variants would have been discovered by the traditional genetic sequencing approach used by most clinics today, which is next generation sequencing on a large gene panel. The only exception would be if one of the variants is actually intronic, as stated in the abstract, but I don't think any of them are intronic and this may have been an error (see point #7 above). However, the traditional NGS approach would not give you information about pathogenicity of the variants and the GWAS approach provided evidence for pathogenicity of the G843E variant. In my opinion, the strongest evidence here for pathogenicity of G843E is the functional data in your zebrafish model. But the GWAS results provide yet another piece of evidence.

11. In the Introduction, lines 88-89, you state that up to 70% of RP cases in Japan have negative or inconclusive results on "targeted re-sequencing." That is quite high. What exactly is targeted re-sequencing (see Question #5)? Is this different from the genetic testing approach that many use (next generation sequencing on a large retinal disease gene panel)? Were any of the patients in your cohort tested with NGS on a large gene panel?

12. In lines 101-102, it almost makes it sound like you are comparing RP with EYS mutations to a "complex disease," and I want to be careful here, because even in RP with reduced penetrance, it is still much more like a Mendelian disorder than a complex genetic disorder. So I think your sentences there are fine, but then I would take your sentence from lines 110-111 ("However, GWAS has never been used to directly search for genetic risks in rare "monogenic" diseases, and its usefulness in such purposes remains unknown."), and I would move it to right after your sentence in 102 (which ends in "monogenic" diseases. That makes it clear that you're saying what GWAS has been used for in the past, and now you are using it in a new way, to look for genetic risks in a rare monogenic disease.

13. In line 105, I'm not sure what is meant by "preferentially in the order of disease contribution." Please clarify or remove.

14. Please explain in the first section of the Results, why were the total DNA samples split into two groups for two separate GWAS studies? How did you decide how to split them? Why didn't you just do one GWAS study instead of meta-GWAS? Was the first group collected first and then a second group collected later and couldn't be added to the first?

15. In line 131 you say that EYS was the only locus was significant, but then you say in line 133 that you found 12 other significant peaks without known RP genes. In line 131, did you mean to say that EYS was the only known retinal disease gene locus that was significant?

16. It would be helpful to analyze these variants in silico in either Polyphen or SIFT or both and say what they are predicted to be (benign or damaging). This information would probably fit best in your discussion of pathogenicity of each of the variants in the results section (lines 144-150).

17. You frequently refer to these 3 variants as the "lead" variants. Were there other variants in LD that you don't discuss? Why not- how did you choose the "leads."

18. In line 169, say reduced penetrant rather than "nonpenetrant."

19. In line 171, please say the vast majority of known pathogenic mutations, because intronic or noncoding regulatory mutations that aren't in exons or at exon/intron boundaries are not typically looked for, and if they are found in WGS, we don't know how to interpret them. So there may very well be pathogenic variants like this, for example noncoding variants that change binding of an enhancer, we just don't know about them. So just add the word known in line 71, and then at the end of that section, line 192, I would add that these results do not rule out a noncoding variant that changes regulation of transcription.

20. Line 172, change slice to splice

21. Line 195, add "the" before "EYS gene."

22. Line 230, "Peak 2 may to confer a distinct pathomechanism." I am not sure I understand this sentence. Please clarify or delete. Perhaps you could say that "Given the high allele frequency and lack of functional data, the pathogenicity of the S2556C variant remains unclear."

23. In line 301, after "oligogenic" I would say "or genetic modifier." To read, "oligogenic or genetic modifier role of EYS."

24. In the discussion, you do a good job discussing the first variant. With regards to the second variant (S2556C), this is less likely to be disease causing given the high AF and the likely benign prediction in ClinVar. I like your comment in lines 313-314 that it may be linked to noncoding variant that was not detected after "targeted re-sequencing" (although I'm still not clear on what this is- see question #5). Your comment in lines 317-319 is less likely, given that RP is a rare disease. You might say that after that sentence, just add "although this is less likely given that RP is a rare disease." You can add that this variant may act in an oligogenic fashion in combination with mutations in other genes; in fact, this might be what you are trying to say in lines 317-319. But I don't think that there are some cases of RP that are actually "complex" genetic diseases in the way that AMD is (with multiple common loci contributing), and I want to avoid having you imply that.

25. I think you should discuss that the third variant (S1653Kfs) was already known to be pathogenic, so it is not surprising that this variant was linked to a peak in GWAS. However, it may be somewhat surprising that no other known pathogenic variants in EYS were found with GWAS, and no other known retinal disease genes were found in GWAS. Why was it just EYS? Please comment on this surprising aspect of the study (surprising to me; maybe there's an explanation), and either offer an explanation or just say you don't know why other mutations or other genes were not found.

Reviewer #3 (Remarks to the Author):

Introduction:

Line 113: Please describe what is meant by presumed ARRP?

Results:

Lines 118-119: Please define what is considered "consistent with AR mode of inheritance" (all definitions in addition to no familial history)

The number of DNA samples in each GWAS study is confusing. In Line 118, the number 944 is stated, but then ultimately it appears GWSAS1 was 432 case and 603 control and GWAS2 was 208 case and 287 control. It is not clear, at least how it is currently written in the text, how the authors went from 944 to these final numbers.

I could not find a location in the Results section of the text that clearly states the location of the Eys variants identified (ie. G843E) in relation to the gene structure. Based on Fig 2A and the legend, it appears that G843 is in exon 16, but this information needs to be clearly stated in the main text.

Figure 2 data and Line 185: The Sanger sequencing results should be provided showing the G843E variant is obtained on sequencing.

Regarding data presented in Figure 2 and also Fig S3: I am confused about what is being presented here. Figure S3 and also the corresponding methods section indicate CRISPR/gRNA editing was performed, but based on the main text and data presented in Figure 2, I was under the impression that the only manipulation done was to place the CAGp to drive the patient's genomic Eys gene expression? Or was Cas9 a part of that method? But, even the legend of S3 makes it seem that Eys was actually edited in these cells? Please clarify.

Figure S2 should be moved to main Figures. Also in Fig S2, it would be helpful to color code the column corresponding to G843 for reader's benefit.

Figure 3 and the experiments/data from zebrafish:

3A: It appears that there is Eys staining also in the inner nuclear and inner plexiform layer. This is

also apparent in the sections from embryonic zfish retinas. Is this an expected location of Eys? "PRC" in 3A is not defined in the figure legend.

There are some major issues with the Morpholino (MO) experiments in zebrafish. First, 7 dpf (the time of sampling in this study) is beyond the generally max ~3 dpf timeframe of blastomere injected morpholino and mRNA efficacy. Second, it does not appear that the authors followed current acceptable guidelines for MO use in zebrafish, which has become an area of concern.

Importantly, see:

D YR Stanier et al., Guidelines for morpholino use in zebrafish. 2017. Plos Genetics.

In particular, phenocopy of a mutant is ideal, but at a minimum MO rescue with zebrafish Eys should be demonstrated, and titrated effects of the MO. (Although this still has the caveat of the 7 dpf sampling time mentioned above). Based on ZFIN.org information, there appears to be several Eys mutants available from ZIRC or EZIRC obtained from the Sanger mutagenesis project. Based on the Methods section, for the rescue experiments shown in Fig 3 using human Eys mRNA, different doses of MO were used in MO alone and MO+mRNA embryos, but this is not justified. The antibody used to label zfish Eys appears to also recognize human Eys (based on the suppliers product information). In this case, then the authors should also demonstrate rescue with zfish Eys mRNA as well as the human Eys mRNAs, to show that the protein is restored and correctly localized.

For the quantifications presented in Fig 3K, the sample size of n=5 or 6 is stated in the figure legend. How many cryosections were analyzed per embryo? At a minimum, 3-4 non-adjacent cryosections should be quantified. In addition, the sample size (actual n examined) would should be increased.

What is the significance of rhodopsin mislocalization found in Fig 3G-J? Was this expected? Is this seen in RP or models of RP??

Finally, it is not clear that a t-test is appropriate statistical test for data in 3K. This would require that the data in each group are normally distributed with equal variances. After that is determined, then either a t-test is justified, or an appropriate non-parametric test should be selected.

Other comments:

There is a large amount of data and information buried in supplemental files. The manuscript would benefit from making at least some of this information more prominent in the main figures/tables.

It seems that the zebrafish studies might be better suited to a separate manuscript as they have several areas of concern as presented, and need to be more fully developed to be of high significance. As mentioned above, several Eys mutants are available from ZIRC or EZIRC that could provide more solid findings in a zebrafish model. In line with this, the authors did not even discuss the data presented in Figure 3 in the Discussion section of the paper.

Reviewers' Comments:

Reviewer #1 (Remarks to the Author):

In this paper the authors use GWAS to identify three association peaks within the EYS gene and study their causality and population impact. The approach is innovative, but might not be generally applicable. The results are believable, the analysis is generally cogent, and the Discussion is informative and complete. The most important contributions of this work are the functional studies of the B843E mutant in zebrafish, and the application of a population genetic approach and segregation analysis to show that it is actually probably causative, both of which are very interesting. Specific comments follow:

RESPONSE:

We are very grateful to the reviewer for considering highly of our work. Since we think that this combinatory approach is useful in detecting founder mutations in Japanese RP and that the sensitivity could be increased by boosting the number of cases and controls to be analyzed in future studies.

COMMENT #1:

Abstract, lines 63-66, "Two were each tagged by a low frequency variant (allele frequency < 0.05); a known founder Mendelian mutation (c.4957dupA, p.S1653Kfs*2) and a presumably hypomorphic non-synonymous variant (c.2528G>A, p.G843E)." This sentence is a little unclear--are c.4957dupA and c.2528G>A the two tagged variant? If so, moving them to immediately behind Two might clarify it ('Two, a known founder...').

RESPONSE:

We apologize for the confusion. We wanted to say that the 2 of the 3 "peaks" detected by GWAS was each in LD with (tagged by) c.4957dupA, and c.2528G>A. Therefore, the sentence was changed as follows at P4, lines 63-67.

"Each of the two were in linkage disequilibrium (LD) with a different low frequency variant (allele frequency < 0.05); a known founder Mendelian mutation (c.4957dupA, p.S1653Kfs*2) and a presumably hypomorphic non-synonymous variant (c.2528G>A, p.G843E). c.2528G>A newly solved 7.0% of Japanese ARRP cases....."

COMMENT #2:

Introduction, lines 93-95, "Furthermore, there is a growing interest in the detection of prevalent founder mutations, which are also potential targets of AAV-mediated genome-editing therapy.": If there is a particular reason founder mutations would be good targets for AAV mediated therapy, the authors should probably state it, because it is not clear to the average reader.

RESPONSE:

Thank you for pointing this out. We have modified the sentence to make our intention clear, which now reads;

"Furthermore, there is a growing interest in the detection of prevalent founder mutations as they may serve as potential candidates of mutation-specific therapy including genome-editing therapy that targets a small specific area of the genome. Therefore, its best application would be founder mutations found in a large number of patients."

COMMENT #3:

3, Results, Detection..., p. 7, line 118, "We gathered a total of 944 DNA samples from unrelated patients...": The authors should include a bit more description of the patient group regarding ascertainment centers and inclusion criteria and comparison of the demographics of the two groups and controls.

RESPONSE:

Thank you for the suggestion. But note, the size of entire Japan with relatively a homogenous population is smaller than that of California. We have expanded the description in the Results and further prompted readers to Methods section for details at P7, lines 122-129, which reads;

"We gathered a total of 944 DNA samples from unrelated Japanese patients who had been diagnosed RP consistent with the AR mode of inheritance that typically had at least one affected sibling and no affected members in other generations. In addition, isolated cases with no family history as well as an offspring of consanguineous parents were also included. Control samples comprised 924 Japanese individuals. Most of them had been confirmed to have normal fundus through ocular exam. The cases were from northeastern, central, or southern Japan, whereas the controls were mostly from the northeast (see Methods section

for detail).”

And also in the Methods section (P25, lines 445-449)

“Of the cases and controls actually used in meta-GWAS, 284, 284, and 72 cases used were recruited via Tohoku University (northeastern region), Kyushu University (southern region), and Nagoya University (central region), respectively, whereas 797 and 93 controls were from Tohoku University and unknown region (purchased as a normal Japanese DNA set), respectively.”

COMMENT #4:

Results, p. 8, lines 128-129, "The results of the two GWASs are summarized in Figure S1 and Tables S2 and S3.": The authors need to provide some additional detail on the two initial GWAS screens including a summary of the actual genotypes, the test performed including the model tested (e.g., recessive, genomic, a simple allelic test--most likely, etc.), and the type of correction carried out for multiple testing. From just looking at the data from the first two GWAS, it would appear that they would be suggestive rather than significant if a Bonferroni correction were applied. Is this the case? Or are the authors correcting only for the SNPs actually genotyped, since the remainder were imputed and thus were not independent estimates?

RESPONSE:

We are sorry for the lack of the detailed explanation. The association between each SNP and the disease (RP) was modeled using logistic regression with an allele dosage effect and adjusted for the 10 principle component scores. We added the additional description in Methods at P24, lines 427-429, which reads,

“The association between each SNP and RP was modeled as logistic regression with an allele dosage effect and adjusted for the 10 principle component scores. The Wald Test was used to determine the significance of association for each SNP.”

We did not apply multiple testing correction (FWER control) such as Bonferroni correction for the imputed SNPs. Due to the linkage disequilibrium (LD), Bonferroni correction is known to be too conservative when large numbers of densely spaced SNPs are evaluated for association with traits. The significance level (α) of 5.0×10^{-8} , proposed by Risch N and Merikangas K (*Science* 273, 1516-1517 (1996)), is widely used in GWASs, and some

simulation studies based on real data (i.e imputed GWAS or genome sequence data) have actually led to similar significance levels with 5.0×10^{-8} , Hoggart CJ et al. (*Genet. Epidemiol.* 32, 179-185 (2008)), Dudbridge F and Gusnanto A (*Genet. Epidemiol.* 32, 227-234 (2008)), Pe'er I et al. (*Genet. Epidemiol.* 32, 381-385 (2008))). Therefore, we adopted 5.0×10^{-8} as the significance level and 1.0×10^{-5} as the suggestive level of GWAS significance. Since the second GWAS was performed for replication and meta-analysis, it is not considered necessary for the second GWAS to reach the genome-wide significance level alone.

COMMENT #5:

Results, p. 8, lines 134-136, "Subsequent conditional analysis of the EYS locus detected 3 independent genome-wide significant signals ($P < 5.0 \times 10^{-8}$; Peaks 1 - 3 in the order of significance, Table 1 and Figure 1B)": The authors could expand the explanation of how this was done--what constituted the conditional analysis and how it differed (and improved) the initial analysis.

RESPONSE:

Thank you for your suggestion. Stepwise conditional analysis has been used as a tool to identify secondary association signals at a locus (e.g. Lango Allen H et al. *Nature* 467, 832-838 (2010), Ripke S et al. *Nat. Genet.* 43, 969-976 (2011), Sklar P. et al. *Nat. Genet.* 43, 977-983 (2011)). We should have cited the references and written the meaning of the analysis. In accordance with the reviewer's comment, we expanded the explanation and revised following points:

In Results P8, lines 144-146, we added a sentence;

"To investigate whether the EYS locus contains multiple variants independently associated with the disease that are not included in the same linkage disequilibrium block, we performed conditional analysis for the EYS locus."

In Methods at P26, lines 451-456, we added a sentence;

"Stepwise conditional analysis has been used as a tool to identify secondary association signals at a locus⁴⁶. ...These steps of adding the variants dosages to the covariate one by one were repeated until there were no variants satisfying the genome-wide significance level ($P < 5.0 \times 10^{-8}$). "

COMMENT #6:

Results, p. 9, lines 149-150, "...registered as benign/likely benign in the ClinVar (<https://www.ncbi.nlm.nih.gov/clinvar/>).": How was this change evaluated by the other commonly used estimators (Polyphen2, Mutation Taster, SIFT, etc.)? Also, what is the frequency of homozygotes in the Japanese population? If it is in HWE and has a high penetrance as expected for Mendelian diseases, homozygotes for this change would give too many cases by an order of magnitude. If it is hypomorphic, is there any difference in the phenotype of patients with this variant (e.g., age of onset, rate of progression, etc.)?

RESPONSE:

In response to the COMMENT, we have inserted predicted function of G843E, S2556C, and S1653Kfs estimated using PolyPhen2, SIFT, Mutation Taster, and CADD in new Table 2.

The frequency of the homozygotes G843E, S2556C, and S1653Kfs was 0.0%, 0.0% and 4.6% as shown in the Table 3.

We completely agree with the reviewer in that the number of observed G843E homozygotes among cases is much lower than expected, which has been discussed (P18, lines 319-320). We also agree with the reviewer that it is important to compare the phenotypes between G843E carriers versus non-carriers among patients with biallelic *EYS* mutations. However, at a glance, the difference was not obvious. Then, we noticed that this is not simple because those with milder phenotype with G843E patients may be asymptomatic (as with the case of asymptomatic brother of YWC133; see Figure S3 for detail) and are unlikely to come to ophthalmologists to begin with, thus the comparison would be largely biased. This is probably the precise reason why we see very few G843E homozygotes compared to the expectation and the phenotypes of the homozygotes were not necessarily milder (data not shown).

A sentence was added to explain this at P19, lines 326-332, which reads;

"Unfortunately, a reliable phenotypic comparison between G843E carriers versus non-carriers among patients with biallelic *EYS* mutations was not possible because many patients with G843E and milder phenotype are unlikely to be included in the genetic analysis to begin with. This is supported by the presence of asymptomatic ARRP patient with G843E and S1653Kfs mutations (the brother of YWC133), who were erroneously assigned as unaffected prior to a thorough ocular examination. This could explain the disproportionately small number

of G843E homozygotes relative to the expectation. “

COMMENT #7:

Results, p. 8, lines 140-144, "Peak 1 (rs76960013, AF = 0.0414, odds ratio (OR) = 3.95, P = 1.18×10^{-13}) was in linkage disequilibrium (LD; $r^2 = 0.68$) with c.2528G>A (p.G843E; hereafter termed G843E; Table 1). G843E with an AF (0.0171) unusually high for ARRP has been described in conflicting ways in past reports, as having uncertain significance²³, being non-pathogenic²⁴, possibly being pathogenic...": The authors are correct that this also seems very high, although they deal with this well later on. Is this the case in published screens? Similarly, are the cases associated with this variant distinctive relative to the rest, especially regarding severity of the RP? Can you expand the analysis of family YWC133 to additional patients as this would be highly convincing. Are there any clinical data on the 12 G834E homozygotes with arRP in Table 2? One might expect them to be milder, with later onset and slower progression, etc.

RESPONSE:

In the two major genetic screening study of Japanese RP patients published to date (Refs 11 and 13), both do not touch upon G843E. This is because the cut-off allele frequency for recessive variants were set at < 0.5% in both studies, which is much lower than the allele frequency of G843E (1.71% from ToMMo database comprising whole genome sequence data of 4,773 normal Japanese).

We did take a look at the available clinical data of patients with biallelic *EYS* mutations and compared the phenotypes between those with and without G843E mutations and found that difference was unexpectedly not very obvious. For example, one of the G843E homozygotes is young (37 years old), yet has typical pigmentary degeneration with visual field constricted within 10 degrees bilaterally. Although it may seem to be contradictory to G843E being hypomorphic, as mentioned above (response to COMMENT #6), we now think this is because patients with G843E and milder phenotype are unlikely to be included in the genetic analysis to begin with. Nevertheless, among the 66 patients with biallelic *EYS* mutations for which clinical data was immediately available, we found that average age of 31 patients with G843E was older compared to those 35 patients without (61.6 YO versus 53.4 YO, P = 0.042

unpaired T-test) with no difference between the visual acuity in both groups. This is consistent with G843E yielding milder phenotype. However, we think it is beyond the scope of this paper to further extend the clinical analysis to tease out small differences between the two groups. We think it should be published as a separate clinical paper following a meticulous analysis. Unfortunately, as for other family members of YWC133, we have no access.

COMMENT #8:

Results, p. 9, lines 152-153, "It was detected even after removing solved cases with biallelic EYS mutations...": What do the authors mean by 'detected'? rs79476654 by itself only gave a $p = 2.54 \times 10^{-8}$, which is barely significant after correction.

RESPONSE:

We have modified the sentence as follows;

"It remained statistically significant even after removing solved cases....."

COMMENT #9:

Results, p. 9, general and Table 1: Were the 3 SNP association peaks in linkage disequilibrium with each other? Fig. 1C is helpful, but an explicit test would be nice to see.

RESPONSE:

The 3 SNPs were not in LD with each other. If they are included in LD each other, the Peak 2 and 3 must disappear after the conditional analysis. In order to provide an answer to the query, the LD plot in Fig.1C has been modified to show the relationships between each peak SNP. The details of r^2 were 0.003, 0.013 and 0.013 for Peaks 1 and 2, Peaks 1 and 3, and Peaks 2 and 3, respectively.

COMMENT #10:

Table 1: It would help to see the p values for the linked variants directly. They should, in theory, be as high or higher than the corresponding associated SNPs.

RESPONSE:

Thank you for your suggestion. In accordance with the reviewer's comment, we added the P-values for the G843E and S2556C in Table 1. The P-value of S1653Kfs was not available

because the variant was not included in the imputed genotypes of the GWAS analysis. As you may expect, the *P*-values of the non-synonymous variants were less significant compared to the lead SNPs in LD.

COMMENT #11:

Results, Detection..., general: It would be helpful for the authors to expand the description of the association studies of each of the putatively associated SNPs to a haplotype of high LD SNPs, as this would help to argue against spurious association. Association of the causative variation with a SNP also implies a founder effect, which could be verified by identifying an associated haplotype. Also, if the 3 associated SNPs are not in LD, or if they are part of separate risk haplotypes, as appears from Fig. 1C, it would help argue for identification of 3 independent and separate loci, even if they are within the same gene.

RESPONSE:

As stated in the reply to COMMENT #9, the lead variant for each of the 3 signals detected by GWAS were in a different non-overlapping LD blocks. This means that the associated haplotypes are also independent. In addition, each peak had only one of the nonsynonymous variants (G843E, S2556C or S1653Kfs) in LD under the condition of $r^2 > 0.6$. Therefore, it could be considered that the pair variants in LD (e.g. Peak1 and G843E) are each part of a unique haplotype.

COMMENT #12:

Results, Expression..., p. 11, lines 189-192, "These results favor against the presence of an intronic mutation in LD with Peak 1 that results in altered splicing and a presumed premature termination of the reading frame, but support G843E as the causal mutation linked to Peak 1.": These results do argue against this change causing a change in splicing, but do not address a change in expression, or decay due to an associated extraneous sequence change. Did the authors consider the possibility of the G843E change causing decreased expression through alteration of codon-usage?

RESPONSE:

Since vast majority of pathogenic *EYS* mutations are null alleles we considered that pathogenic allele would most likely result in qualitative changes of mRNA. However, if

hypomorphic allele is to be considered, we agree with the reviewer that we should also consider reduced expression. Meanwhile, the mechanism by which G843E causes dysfunction of EYS has not been investigated in this study. A sentence has been added to address this at P12, lines 205-206, which reads;

“Meanwhile, the presence of a noncoding variant that causes the disease through changes in regulation of *EYS* transcription cannot be ruled out.”

COMMENT #13:

Results, Expression..., p. 11, lines 181-192, ...transcription of EYS...Peak1." This paragraph, and its relation to Fig.s 2 and S3, are a little unclear. First, please define LCL at its first use. Do untreated LCLs not express EYS mRNA as it would appear from Fig. 2B? Demethylation appears sufficient for expression of exons 4-5 and 14-18, but not 40-43 in wt and G843E cells, is this correct? If so, why? If the S1653fs mutation undergoes NMD, as suggested by emetine treatment, why are exons 4-5 and 14-18f detected without treatment? Is there a small amount of mRNA that escapes decay or undergoes an alternate splicing pathway, or is Tv6-3 or 7 perhaps expressed in the lens? Finally, Fig. S3C-F, explaining replacement of the S1653fs mutation, do not appear to be referenced in the text.

RESPONSE:

We regret that we should have explained better.

Yes, EYS mRNA was undetectable in untreated LCLs. And yes, demethylation was sufficient for expression of exons 4-5 and 14-18, but not exons 40-43 in LCLs. This is best explained by the differential expression of distinct EYS isoforms as suggested by the reviewer. Unfortunately, the reasons for the differential response to demethylating agent and NMD inhibitor is unknown at this point. One possibility is that the longest mRNA isoforms (Tv1 and 4) of more than 10kb require robust transcription initiation compared to other much shorter isoforms.

We have added a sentence in Figure 2 legend to explain the differential expression of long and shorter isoforms at P43, lines 822-823, which reads;

“Meanwhile, mRNA for exons 4-5 and 14-18 were detectable, possibly reflecting the differential expression of distinct *EYS* isoforms.”

As for Figs S3C-F, thank you for it pointing out. We have improved the description on methods for “mRNA analysis using patient-derived lymphoblastoid cell lines (LCLs)” at P27-29, so that each and all of the Figure panels are now cited in the text.

COMMENT #14:

Results, Functional..., p. 12, lines 201-202, "During development, Eys expression was observed after 4 days post-fertilization (dpf, Figure 3D-F).": It looks like there might be some expression at 3 dpf. Is this the case? Did the authors do a quantitative test?

RESPONSE:

Thank you for the comments. As you pointed out there were potentially weak signals on IHC at 3 dpf, However, they were not related the cilium that they should. So, we could not conclude that the weak signals observed at 3 dpf represent true expression. In the absence of convincing IHC data, we did not perform quantitative analysis such as RT-PCR. However, because the phenotype of *ey*s morphant is apparent in later stages (4-7 dpf), we focused on 4 to 7 dpf in the downstream experiments.

COMMENT #15:

Results, Functional..., p. 11-12, general: While the results obtained with morpholino knockdown and rescue are convincing to this reviewer, they do not meet the current guidelines for morpholino use in zebrafish (Didier et al., PLoS Genetics, 2017), which would require knockout confirmation. The authors should address this either by doing the knockout and rescue or justifying not doing so in the Discussion.

RESPONSE:

It is quite an important point. The morphants in our experiment had shown the mislocalization of rhodopsin, and a zygotic mutant (knockout of *ey*s) also shows same phenotype (Yu M. et al. Biol Open. 2016 15:1662-1673, Reference 29). Moreover, as the wildtype *ey*s mRNA co-injection had rescued the phenotype, we concluded that the morpholinos had given the *ey*s knockdown-specific phenotype. However, as you pointed out the efficacy and specificity of morpholino knock down should be checked more carefully.

In the revision, we tried tested three different splice site morpholinos (SPMO1-3). All of them showed the suppression of *ey*s expression in RT-PCR analyses, and gave the phenotype of

rhodopsin mislocalization at 7 dpf. With these lines of evidence combined, we believe that our experiment met the guideline that you showed.

COMMENT #16:

Results, Enrichment..., p. 13, lines 219-220, "This establishes that the G843E allele contributes to RP in trans with another EYS mutation, as in ARRP.": The authors might consider replacing 'establishes' with 'strongly suggests' or something similar.

RESPONSE:

This has been corrected accordingly.

COMMENT #17:

Results, Enrichment..., p. 13, line 230, "Meanwhile, Peak 2 may to confer a distinct pathomechanism.": This needs to be explained and expanded.

RESPONSE:

We have toned down the sentence by replacing "distinct" with "different" and directed the reader to Discussion where the topic is dealt, which now reads;

"Meanwhile, Peak 2 may to confer a different pathomechanism (see Discussion)."

COMMENT #18:

Results, Enrichment, p. 14, lines 247-255, "None of the unaffected...mild form of RP": It would really strengthen the manuscript to show the detailed clinical data from this patient, and any others with homozygous or heterozygous G843E mutations.

RESPONSE:

At the request of the reviewer, we have now added a new supplementary Figure (Figure S3) summarizing the phenotype of the asymptomatic affected brother of YWC133 with G843E and S1653Kfs mutations. Since the phenotypes of patients with homozygous or heterozygous G843E mutations also included severe cases, it was difficult to show a "representative" case. Instead we displayed a representative case of S1653Kfs homozygotes,

which uniformly showed severe retinal degeneration phenotype in the elderly.

COMMENT #19:

Discussion, p. 16, lines 270-271, "This confirmed the quality of GWAS and its ability to effectively detect classical Mendelian mutations.": This is a little unclear from the results presented. While the GWAS was useful in that it pointed to 3 variants, each of these variants would have been identified (or perhaps were identified previously) by WES or WGS. The authors might wish to soften this statement. They might also mention that the use of GWAS is dependent on there being a common founder for each mutation, so that this will not work in all cases, especially if there are multiple independent mutations causing the RP.

RESPONSE:

Thank you for the important suggestions. We agree with the reviewer and modified the sentences accordingly at P17, lines 301-303, which reads;

"This confirmed the quality of GWAS and its ability to effectively detect classical Mendelian mutations, although S1653Kfs could have been identified with sequencing alone in this case. At the same time, successful application of GWAS is dependent on there being common founders, which may limit its use in a highly heterogenous population."

COMMENT #20:

Discussion, p. 19, lines 325-329, "In conclusion, this study provides a novel GWAS-based framework for systematically detecting disease-associated variants, unbiased with regard to genomic location and mode of genetic influence, in so-called "monogenic" disorders. It also highlights the under-appreciated significance of non-Mendelian high frequency variants that may significantly account for the undetermined heritability of various inherited diseases.": It is actually unclear from the results presented here that the GWAS was necessary for the most important findings presented in this paper, which are the roles of the p.S1653Kfs*2 and especially the p.G843E mutation in their series of patients. These were detected in their sequencing work, perhaps before the GWAS was performed, and had been reported as founder mutation and suspect mutation before. In addition, one of these variants is a classical high-penetrance Mendelian allele, while the second is still Mendelian and monogenic, but simply has reduced penetrance. These conclusions need to be revised a bit, perhaps

acknowledging the usefulness of the association results in identifying 3 discrete regions within EYS and also mentioning the importance of a founder effect for application of association studies using non-causative markers.

RESPONSE:

We have rearranged the concluding paragraph taking the comment into consideration at P22, lines 379-385, which reads;

“In conclusion, this study demonstrates the usefulness of GWAS in identifying disease-associated loci, in so-called “monogenic” disorders, which is dependent on the presence of founder mutations. It also highlights the under-appreciated significance of high frequency variants that may significantly account for the undetermined heritability of various inherited diseases. At the same time, significance of the identified variants may extend beyond genetic diagnosis as they may simultaneously serve as ideal targets of local genome treatments.”

COMMENT #21: Suggestions for English grammar and usage:

1. 84 "presenting with a various hereditary pattern": maybe 'variety of hereditary patterns'
2. 99 "Genome-wide association study (GWAS) is a type of analysis": maybe '...studies are a...".
3. 189 "These results favor against...": Maybe 'argue against' or 'favor causality of the G843E change over...' or something similar.
4. 195 Among mammals, only primates have EYS gene.: Maybe 'have the EYS gene' or 'have EYS'.
5. 201 "...photoreceptors in adult fishes (Figure 3A, B).": Maybe '...adult fish...'. 'Fishes' is usually used to refer to multiple species of fish, while the plural of fish is usually fish.
6. 302 "among carriers of EYS-G843E, but this maybe attributable": Maybe 'may be' instead of 'maybe'.
7. 321 "finding stress the importance": Maybe 'findings stress' and change '...and emphasize...'.

RESPONSE:

Thank you for pointing out a number of grammatical errors. All the suggested changes have been addressed.

Reviewer #2 (Remarks to the Author):

THE MAIN COMMENTS;

This is a unique, well written, and interesting paper in which the authors use GWAS in a cohort of patients with RP in which the genetic etiology of disease has not been found. GWAS revealed variants in EYS, at least 2 of which helped explain disease in a subset of the patients. What remains unclear is what sequencing was performed in these patients prior to GWAS (the term “targeted re-sequencing” is mentioned several times throughout the manuscript, but I can’t see exactly what this is. What genes were sequenced and why was it “re-sequencing?” Were they previously sequenced before the re-sequencing? Was NGS panel testing performed at any point?), and why was EYS the only gene that was found on GWAS out of the 80 plus genes that are known to cause RP?

RESPONSE:

Thank you very much for the overall favorable view on our work. At the same time, we regret that the use of the term “targeted re-sequencing” has brought about a confusion. We have sequenced the all the coding exons and immediate intronic boundary of the known 83 RP genes (registered as of September 2017 in the RETNET) with NGS and it is virtually the same as “NGS panel testing”. Therefore, we have replaced “targeted re-sequencing” with “NGS panel testing” throughout the manuscript.

The main takeaway from this paper is that the G843E variant, previously of unknown significance, now has good evidence for pathogenicity, including functional data in zebrafish and segregation data, in addition to the GWAS evidence. The pathogenicity of S2556C remains unclear, and S1653Kfs was already known to be pathogenic. It is interesting that GWAS can be used to help provide evidence for pathogenicity of variants, but the authors need to address why they think EYS was the only gene found, and whether that raises any concerns. For example, do they think EYS is by far the most common cause of ARRP in unsolved cases? Relevant to this question is what genetic testing they already had (again,

what was the "targeted re-sequencing" they had?).

In addition, several point by point comments are below. I have given fairly extensive comments and revisions, but overall I recommend publication of this unique paper.

RESPONSE:

We are again very grateful for the insightful comment. As the reviewer pointed out, we think that the main reason that only *EYS* was detected was because *EYS* mutations is by far the most common disease gene accounting for a large number of unsolved cases. For example, we found that among the 848 (out of 1,204) RP patients unsolved after NGS panel testing, approximately 25% had heterozygous deleterious mutation in *EYS* (see Figure 2, reference __, Koyanagi et al., J Med Genet 2019). Also the number of cases and controls is a very important factor in GWAS. We have added a few sentences to discuss this at P17, lines 305-313, which reads;

“Another important factor appears to be the study size as with the case of GWAS for common traits. While there are a few founder ARRP mutations including those in *EYS*, *USH2A*, *RP1*, *SAG*, and *RP1L1* reported in Japanese population, those in *EYS* is by far the most frequent. It is likely that this limited the GWAS to detect only exceedingly frequent founder mutations in *EYS*. However, increasing the number of cases and controls should greatly facilitate detection of less frequent founder mutations as demonstrated in many recent large-scale multicenter GWAS studies that has boosted the number of disease- associated loci from a few initially to often dozens including those for glaucoma and age-related macular degeneration”

COMMENT #1:

Abstract line 60, uncover “the” genetic basis

RESPONSE:

The sentence has been modified accordingly.

COMMENT #2:

Abstract line 61, applied “a” 2-step genome-wide association

RESPONSE:

The sentence has been modified accordingly.

COMMENT #3:

Abstract line 65, I would say “possibly” hypomorphic instead of “presumably” hypomorphic. Later you present evidence for why you think it is hypomorphic, but here in the abstract it is unclear why you are presuming it to be hypomorphic.

RESPONSE:

Thank you for the suggestion. The sentence has been modified accordingly.

COMMENT #4:

In the abstract, line 68, I would say “possibly” not likely, as will be discussed in the points below. Given high AF and lack of functional data or segregation data like you have for the G843E, I think the role of this variant in disease is still very unclear.

RESPONSE:

We agree with the reviewer suggestion to be conservative about the GWAS results with no functional analysis. The sentence has been modified accordingly.

COMMENT #5:

Introduction line 79, what is meant by “simplex sequence-based approach?” Please reword or remove. Do you mean the “traditional next-generation sequencing approach?”

RESPONSE:

In response to the comment, “simplex sequence-based approach” has been replaced with “next-generation sequencing approach”.

COMMENT #6:

What is “targeted re-sequencing.” Is this sanger sequencing of specific genes? Is it just exons? Exons plus exon-intron boundaries? The entire gene? And why “re”? Was this gene

sequenced in these patients previously?

RESPONSE:

“Targeted re-sequencing” is same as “NGS panel testing”. We are sorry for bringing about confusion. For targeted re-sequencing, we have sequenced the all the coding exons and the flanking immediate exon-intron boundaries of the known 83 RP genes with NGS. We have replaced “targeted re-sequencing” with “NGS panel test” throughout the manuscript. and Methods (P22-23, lines 395-397) has been expanded to explain this sequencing method.

COMMENT #7:

When you refer to allele frequencies, what database are you using?

RESPONSE:

We appreciate the reviewer for pointing out the missing information. The database used for estimating allele frequency was ToMMo (<https://www.megabank.tohoku.ac.jp/english/>), a genomic database from whole genome sequencing of 4,773 Japanese healthy individuals. We have added this information in the Methods and cited the corresponding reference.

COMMENT #8:

Abstract line 68, you say the third peak is tagged by an intronic common variant, but later when you say the 3 variants, none of them seem to be intronic. 2 are missense mutations (G843E and S2556C) and 1 is a frameshift (S1653Kfs). Please clarify.

RESPONSE:

Thankfully, the reviewer has properly pointed out a mistake. We have corrected the corresponding sentence at P4, lines 68-69, so that it reads;

“The third peak was in LD with a common non-synonymous variant (c.7666A>T, p.S2556C),
....”

COMMENT #9:

Somewhere in the results, I would like to see, for each of the 3 variants, how many of the

total RP cases in the cohort had that variant, and of those, how many were homozygous, how many were heterozygous, and of the heterozygotes how many had a second known pathogenic or likely pathogenic variant in EYS? In other words, if there were 432+208 cases analyzed (640), say 50 of them had the first variant (G843E), and 20 of those were homozygous, and the other 30 were heterozygous. Of the 30 heterozygotes, say 10 had a second known pathogenic or likely pathogenic variant in EYS. That is useful information, because now I know that 30 of those 640 RP cases are potentially explained by EYS mutations. Without those numbers, it makes it harder to make sense of the results.

RESPONSE:

In the original Table 2, we did provide the genotype count information for G843E. Upon reviewer's request, we have revised Table 3 (original Table 2) so that genotype information for the two other variants (S1653Kfs and S2556C) are also displayed.

COMMENT #10:

In the Discussion, you should point out that these variants would have been discovered by the traditional genetic sequencing approach used by most clinics today, which is next generation sequencing on a large gene panel. The only exception would be if one of the variants is actually intronic, as stated in the abstract, but I don't think any of them are intronic and this may have been an error (see point #7 above). However, the traditional NGS approach would not give you information about pathogenicity of the variants and the GWAS approach provided evidence for pathogenicity of the G843E variant. In my opinion, the strongest evidence here for pathogenicity of G843E is the functional data in your zebrafish model. But the GWAS results provide yet another piece of evidence.

RESPONSE:

In principle, we agree with the reviewer. It is obvious that exonic variants can be recognized with panel sequencing that targets all coding exons. However, as for S2556C in LD with Peak 2, since the allele frequency is very high with lower odds ratio, more like a typical risk variant for common disease in which the true pathogenic variant(s) is typically unknown, we are very unsure about the pathogenicity of S2556C itself. This has been explained at P21, lines 371-374, which reads;

“At the same time, it is possible that the unknown true pathogenic variant(s) different from the S2556C variant lies deep in a noncoding region as typical for signals detected by GWAS in common diseases.”

COMMENT #11:

In the Introduction, lines 88-89, you state that up to 70% of RP cases in Japan have negative or inconclusive results on “targeted re-sequencing.” That is quite high. What exactly is targeted re-sequencing (see Question #5)? Is this different from the genetic testing approach that many use (next generation sequencing on a large retinal disease gene panel)? Were any of the patients in your cohort tested with NGS on a large gene panel?

RESPONSE:

Please refer to the RESPONSE for COMMENT #6.

The diagnostic rate is low but it is similar to another large genetic screening of RP which reported a rate of 36.3% (reference 13).

For cases unsolved by NGS panel, we have further performed whole genome sequencing in ~50 cases and whole exome sequencing in over 100 cases. But these tests only increased the diagnosis by a few percent (data not shown).

COMMENT #12:

In lines 101-102, it almost makes it sounds like you are comparing RP with EYS mutations to a “complex disease,” and I want to be careful here, because even in RP with reduced penetrance, it is still much more like a Mendelian disorder than a complex genetic disorder. So I think your sentences there are fine, but then I would take your sentence from lines 110-111 (“However, GWAS has never been used to directly search for genetic risks in rare “monogenic” diseases, and its usefulness in such purposes remains unknown.”), and I would move it to right after your sentence in 102 (which ends in “monogenic” diseases. That makes it clear that you’re saying what GWAS has been used for in the past, and now you are using it in a new way, to look for genetic risks in a rare monogenic disease.

RESPONSE:

We appreciate the great suggestion. The paragraph has been amended according to the

reviewer's suggestion.

COMMENT #13:

In line 105, I'm not sure what is meant by "preferentially in the order of disease contribution."
Please clarify or remove.

RESPONSE:

We have removed the phrase.

COMMENT #14:

Please explain in the first section of the Results, why were the total DNA samples split into two groups for two separate GWAS studies? How did you decide how to split them? Why didn't you just do one GWAS study instead of meta-GWAS? Was the first group collected first and then a second group collected later and couldn't be added to the first?

RESPONSE:

Thank you for alarming us that the description on the study design may be inadequate. The design of two stage was outlined in the supplementary reporting summary in the "In the Life sciences" section under "study design" subheading as follows;

"Since it was very difficult to estimate the outcome initially because we could not find a GWAS targeting recessive Mendelian disorders, we estimated the size of the second GWAS based on the results of the first GWAS assuming that meta-GWAS was to be performed. The first GWAS was carried out using all the samples available at that time. Sample sizes for the second GWAS were calculated so that top 5 signals would reach statistical significance using on-line sample size calculator (<https://www.stat.ubc.ca/>) adopting a two-sided alpha-level of 0.05, 80% power. However, size was eventually restricted by the availability of the samples because the disease studied was a rare disease with a prevalence of 1 in 4000."

These sentences were added to the Methods (P25, lines 438-446).

Another reason why we ended up performing two-stage GWAS instead of combining the data to perform one-stage GWAS is because the version of the DNA chip used was different between the two stages. The chip used for initial GWAS was upgraded from CoreExome-24 V1.1 to V1.2 by the time we performed genotyping for the second group of patients. This

information is already provided in the Methods.

COMMENT #15:

In line 131 you say that EYS was the only locus was significant, but then you say in line 133 that you found 12 other significant peaks without known RP genes. In line 131, did you mean to say that EYS was the only known retinal disease gene locus that was significant?

RESPONSE:

Not exactly. We are sorry our explanation was suboptimal. We detected 1 peak (EYS locus) with genome-wide significance ($P < 5.0 \times 10^{-8}$) and 12 peaks with possible relevance ($P < 5.0 \times 10^{-5}$). Such description based on two different P-value thresholds is commonly adopted in GWAS studies. We have corrected the sentence to make our intention clearer, at P8, lines 142-144, which reads;

“Among signals that did not reach genome-wide significance ($P < 5.0 \times 10^{-8}$), there were 12 other peaks with possible relevance ($P < 1.0 \times 10^{-5}$) in which no known RP genes were included (Table S3).”

COMMENT #16:

It would be helpful to analyze these variants in silico in either Polyphen or SIFT or both and say what they are predicted to be (benign or damaging). This information would probably fit best in your discussion of pathogenicity of each of the variants in the results section (lines 144-150).

RESPONSE:

We received a similar comment by Reviewer 1 (COMMENT #6). We have added the requested information in new Table 2 (originally Table 1).

COMMENT #17:

You frequently refer to these 3 variants as the “lead” variants. Were there other variants in LD that you don’t discuss? Why not- how did you choose the “leads.”

RESPONSE:

For each signal with a genome-wide significance ($P < 5.0 \times 10^{-8}$), a variant with the lowest

P-value was selected as the “lead variant” among many other neighboring SNPs in LD that are also statistically significant. The method follows previous studies (e.g. Lango Allen H et al. *Nature* 467, 832-838 (2010); see also the RESPONSE to COMMENT #5 from Reviewer #1). Note, the variant with the lowest *P*-value is not necessarily directly associated to the disease, and other variants in the LD block could also be the pathogenic variant.

COMMENT #18:

In line 169, say reduced penetrant rather than “nonpenetrant.”

RESPONSE:

We have followed the reviewer’s request.

COMMENT #19:

In line 171, please say the vast majority of known pathogenic mutations, because intronic or noncoding regulatory mutations that aren’t in exons or at exon/intron boundaries are not typically looked for, and if they are found in WGS, we don’t know how to interpret them. So there may very well be pathogenic variants like this, for example noncoding variants that change binding of an enhancer, we just don’t know about them. So just add the word known in line 71, and then at the end of that section, line 192, I would add that these results do not rule out a noncoding variant that changes regulation of transcription.

RESPONSE:

Thank you for the great suggestion. We have made correction accordingly. As a result, the following sentence has been added at P12, lines 205-207.

“Meanwhile, the presence of a noncoding variant that causes the disease through changes in regulation of *EYS* transcription cannot be ruled out.”

COMMENT #20:

Line 172, change slice to splice

RESPONSE:

Sorry for the typo. This has been taken care of.

COMMENT #21:

Line 195, add “the” before “EYS gene.”

RESPONSE:

This has been addressed.

COMMENT #22:

Line 230, “Peak 2 may to confer a distinct pathomechanism.” I am not sure I understand this sentence. Please clarify or delete. Perhaps you could say that “Given the high allele frequency and lack of functional data, the pathogenicity of the S2556C variant remains unclear.”

RESPONSE:

In short, we think Peak 2 may contribute the disease in a non-Mendelian manner as the risk allele frequency is simply too high for autosomal recessive allele. This discussion has been further expanded in the Discussion at P12, lines 205-207.

Here, we have re-directed the reader to the Discussion by modifying the sentence as follows; “Meanwhile, Peak 2 may to confer a different pathomechanism given the high frequency of the pathogenic allele (see Discussion).”

COMMENT #23:

In line 301, after “oligogenic” I would say “or genetic modifier.” To read, “oligogenic or genetic modifier role of EYS.”

RESPONSE:

Thank you for the suggestion. The change has been made.

COMMENT #24:

In the discussion, you do a good job discussing the first variant. With regards to the second variant (S2556C), this is less likely to be disease causing given the high AF and the likely

benign prediction in ClinVar. I like your comment in lines 313-314 that it may be linked to noncoding variant that was not detected after “targeted re-sequencing” (although I’m still not clear on what this is- see question #5). Your comment in lines 317-319 is less likely, given that RP is a rare disease. You might say that after that sentence, just add “although this is less likely given that RP is a rare disease.” You can add that this variant may act in an oligogenic fashion in combination with mutations in other genes; in fact, this might be what you are trying to say in lines 317-319. But I don’t think that there are some cases of RP that are actually “complex” genetic diseases in the way that AMD is (with multiple common loci contributing), and I want to avoid having you imply that.

RESPONSE:

We agree with the reviewer. Yes, we think it is possible that the risk variant for the Peak 2 acts as a modifier or in an oligogenic fashion in combination with mutations in other genes. We have added the phrase “although this is less likely given that RP is a rare disease” followed by an insertion of a sentence “For example, the risk variant may act in an oligogenic fashion or as a disease modifier in combination with mutations in other genes.”

As for S1653Kfs, its detection with GWAS was unexpected as a large number of homozygotes and heterozygotes had been removed before the analysis with prescreening by NGS panel.

COMMENT #25:

I think you should discuss that the third variant (S1653Kfs) was already known to be pathogenic, so it is not surprising that this variant was linked to a peak in GWAS. However, it may be somewhat surprising that no other known pathogenic variants in EYS were found with GWAS, and no other known retinal disease genes were found in GWAS. Why was it just EYS? Please comment on this surprising aspect of the study (surprising to me; maybe there’s an explanation), and either offer an explanation or just say you don’t know why other mutations or other genes were not found.

RESPONSE:

The response to this comment has been provided in the RESPONSE to THE MAIN COMMENT as follows; “we think that the main reason that only EYS was detected was

because EYS mutations is by far the most common disease gene accounting for many unsolved cases. For example, we found that among the 848 RP patients unsolved after NGS panel testing, approximately 25% had heterozygous deleterious mutation in *EYS* (see Figure 2, reference __, Koyanagi et al., J Med Genet 2019).

Reviewer #3 (Remarks to the Author):

Introduction:

COMMENT #1:

Line 113: Please describe what is meant by presumed ARRP?

RESPONSE:

Thank you for the important comment. We admit that the description of the patients was insufficient. We have expanded the description in the first paragraph of the Results at P7, lines 122-125, which reads;

“We gathered a total of 944 DNA samples from unrelated Japanese patients who had been diagnosed RP consistent with the AR mode of inheritance that typically had at least one affected sibling and no affected members in other generations. In addition, isolated cases with no family history as well as an offspring of consanguineous parents were also included.”
In addition, similar description was added to the Methods at P23, lines 397-399.

Results:

COMMENT #2:

Lines 118-119: Please define what is considered “consistent with AR mode of inheritance” (all definitions in addition to no familial history)

RESPONSE:

Please see response to COMMENT #1.

COMMENT #3:

The number of DNA samples in each GWAS study is confusing. In Line 118, the number 944 is stated, but then ultimately it appears GWSAS1 was 432 case and 603 control and GWAS2 was 208 case and 287 control. It is not clear, at least how it is currently written in the text, how the authors went from 944 to these final numbers.

RESPONSE:

The number of samples used for analysis reduced greatly after removing QC failures and solved cases. We have proved the exact numbers of samples removed for each step in the text but perhaps our presentation was not clear enough. We also provided a supplementary Table (Table S1) that summarized related numbers so that the readers could easily follow how the numbers of cases and controls were reduced from 944 and 924 to 640 and 890, respectively. Nevertheless, we have modified the text and Table S1 in hope to make it more understandable. And modified Table S1 has been reassigned to the main Table 1 to facilitate the understanding of the readers.

COMMENT #4:

I could not find a location in the Results section of the text that clearly states the location of the Eys variants identified (ie. G843E) in relation to the gene structure. Based on Fig 2A and the legend, it appears that G843 is in exon 16, but this information needs to be clearly stated in the main text.

RESPONSE:

Upon the request, we have added exon information for G843E at P12, line 199.

COMMENT #5:

Figure 2 data and Line 185: The Sanger sequencing results should be provided showing the G843E variant is obtained on sequencing.

RESPONSE:

We appreciate the important comment. The requested information has been provided as Figure 2C.

COMMENT #6:

Regarding data presented in Figure 2 and also Fig S3: I am confused about what is being presented here. Figure S3 and also the corresponding methods section indicate CRISPR/gRNA editing was performed, but based on the main text and data presented in Figure 2, I was under the impression that the only manipulation done was to place the CAGp to drive the patient's genomic Eys gene expression? Or was Cas9 a part of that method? But, even the legend of S3 makes it seem that Eys was actually edited in these cells? Please clarify.

RESPONSE:

We are sorry about the confusion. Figure 2 comprises two experiments, both of which uses CRISPR-Cas9 system. First, pCAG was inserted immediately upstream of ATG of EYS using genome editing approach in patient-derived LCLs (A-C). Second, we used the LCL from S1653Kfs homozygote to demonstrate the lack of long isoform is due to nonsense-mediated decay, which could be reversed by correcting the mutation by genome editing (D). Figure S4 (originally S3) only shows the information related to genome editing strategy employed in both experiments. We have changed the title of the Figure S4 to "Genome editing strategy to insert CAG promoter or correct S1653Kfs mutation in lymphoblastoid cell lines (LCLs)"

COMMENT #7:

Figure S2 should be moved to main Figures. Also in Fig S2, it would be helpful to color code the column corresponding to G843 for reader's benefit.

RESPONSE:

Thank you for the great suggestion. This has been addressed.

Figure 3 and the experiments/data from zebrafish:**COMMENT #7:**

3A: It appears that there is Eys staining also in the inner nuclear and inner plexiform layer. This is also apparent in the sections from embryonic zfish retinas. Is this an expected location

of Eys?

RESPONSE:

Thank you for the comment. You are right. As reported *eyes* function is related cilia, so we focused on staining around the cilia, although ganglion cell layer and inner nuclear layer also showed some positive signal. However, the function of *eyes* in inner retinal layers is completely unknown.

COMMENT #7:

“PRC” in 3A is not defined in the figure legend.

RESPONSE:

In Figure 4 legend, PRC is defined as photoreceptors.

COMMENT #7:

There are some major issues with the Morpholino (MO) experiments in zebrafish. First, 7 dpf (the time of sampling in this study) is beyond the generally max ~3 dpf timeframe of blastomere injected morpholino and mRNA efficacy. Second, it does not appear that the authors followed current acceptable guidelines for MO use in zebrafish, which has become an area of concern. Importantly, see: D YR Stanier et al., Guidelines for morpholino use in zebrafish. 2017. Plos Genetics.

In particular, phenocopy of a mutant is ideal, but at a minimum MO rescue with zebrafish *eyes* should be demonstrated, and titrated effects of the MO. (Although this still has the caveat of the 7 dpf sampling time mentioned above). Based on ZFIN.org information, there appears to be several *eyes* mutants available from ZIRC or EZIRC obtained from the Sanger mutagenesis project.

RESPONSE:

Thank you for the comment. As you mentioned MOs cannot not suppress the gene expression for a long time. However, it can suppress the expression up until around 5 dpf (Neuron 42, 703-716, 2004). And this refers to gene expression but the induced phenotype could persist for a longer period of time. Second, we admit that we had not completely met the criteria. Thus, we added some experiments to further assess the efficacy of MO. The

morphants in our experiment showed mislocalization of rhodopsin, a phenotype also observed in a zygotic mutant (knockout of *ey3*) (Biol Open. 2016 15:1662-1673, Reference 29). Moreover, since co-injection of the wildtype *ey3* mRNA had rescued the phenotype, we concluded that the morpholinos had given the *ey3* knockdown-specific phenotype. Furthermore, we tested three different splice site morpholinos (SPMO1-3) in adult zebrafish. RT-PCR analyses showed inhibition of *ey3* expression and immunohistochemistry displayed rhodopsin mislocalization with all MOs. Collectively, we believe that our experiment met the guideline that has been proposed by the reviewers.

COMMENT #8:

Based on the Methods section, for the rescue experiments shown in Fig 3 using human *ey3* mRNA, different doses of MO were used in MO alone and MO+mRNA embryos, but this is not justified.

RESPONSE:

We apologize for the unclear statements in Methods section. There was no difference in MO concentrations between groups within a given experiment. However, concentration was not identical between different MOs. Before the phenotypic analysis of *ey3* morphants, we performed pilot injection of each MO in a smaller scale to see the systemic adverse effects. In this pilot study, ATG, SP2, SP3 MO did not show toxicity at 380mM. However, SP1 MO showed some curvature in body axis indicating toxicity. This phenotype disappeared at a lower concentration, i.e., 200mM SP1 MO. Therefore, we combined 200mM SP1MO and 200mM ATGMO. For other MOs, we used 380mM MO with or without mRNA injection.

COMMENT #9:

The antibody used to label zfish *Eys* appears to also recognize human *Eys* (based on the suppliers product information). In this case, then the authors should also demonstrate rescue with zfish *ey3* mRNA as well as the human *ey3* mRNAs, to show that the protein is restored and correctly localized.

RESPONSE:

We have tried visualizing the restored human *ey3* with immunohistochemistry, with little luck so far, perhaps due to the relatively low signal to noise ratio. In addition, the aim of the

experiment is to show the pathogenicity of human G843E mutation, not to show the detailed function of zebrafish eyes. Considering that the effect of exogenous human eyes was clearly demonstrated by assessing rhodopsin mislocalization and that functional assay of G843E already fulfills the guideline for the use of MO in zebrafish, we hope the current data is satisfactory.

COMMENT #10:

For the quantifications presented in Fig 3K, the sample size of n=5 or 6 is stated in the figure legend. How many cryosections were analyzed per embryo? At a minimum, 3-4 non-adjacent cryosections should be quantified. In addition, the sample size (actual n examined) would should be increased.

RESPONSE:

Thank you for pointing out an important issue. We usually arrange around 10 embryos per section and distinguish each other by their position on the section to avoid repeated counting of the same embryo. At least 6 non-adjacent sections per groups are assessed. In the revised manuscript, the sample size has been increased.

COMMENT #11:

What is the significance of rhodopsin mislocalization found in Fig 3G-J? Was this expected? Is this seen in RP or models of RP??

Finally, it is not clear that a t-test is appropriate statistical test for data in 3K. This would require that the data in each group are normally distributed with equal variances. After that is determined, then either a t-test is justified, or an appropriate non-parametric test should be selected.

RESPONSE:

Rhodopsin mislocalization has been reported in mice models of RP (*J Neurosci* 1994, 14:5818, *PNAS*, 2002 99:5698., *PNAS* 2004, 101:16588). In addition, as eyes is localized to the connecting cilium, the mislocalization of rhodopsin caused by defective ciliary transport is a reasonable phenotype. Moreover, we have demonstrated that the mislocalization can cause the photoreceptor cell death (*PLoS One*. 7, e32472, 2012).

With regard to the statistical assessment we completely agree with you. We repeated

experiments and increased N numbers, then analyzed using Wilcoxon rank sum test.

Other comments:

COMMENT #12:

There is a large amount of data and information buried in supplemental files. The manuscript would benefit from making at least some of this information more prominent in the main figures/tables.

RESPONSE:

Thank you very much for the proposal. Taking the proposal into consideration, we have now reassigned Table S1 as Table 1 and Supplementary Figure S2 as Figure 3 so that both of them are now a part of the main Tables/Figures

COMMENT #13:

It seems that the zebrafish studies might be better suited to a separate manuscript as they have several areas of concern as presented, and need to be more fully developed to be of high significance. As mentioned above, several eys mutants are available from ZIRC or EZIRC that could provide more solid findings in a zebrafish model. In line with this, the authors did not even discuss the data presented in Figure 3 in the Discussion section of the paper.

RESPONSE:

The main message of the work is the usefulness of combinatorial GWAS/sequencing approach in screening for high frequency variants in ARRP and thus we wanted to keep the functional analysis of G843E concise particularly because we have other supporting data (Segregation analysis, LCL work, whole genome sequencing, and GWAS). However, we very much like to keep the zebrafish data in this manuscript as it does enhance the quality of the work as indicated by the Editor and the 2 other reviewers. As mentioned above, we have done our best to comply with the guideline by adding a bulk of data so that the data are more interpretable. In addition, thank you for reminding us about not referring to the zebrafish work in the Discussion. We have stressed the importance of the direct functional assessment of

the G843E variant to determine its pathogenicity in the Discussion at P18, lines 316-317, which reads;

“Herein, we provide a direct evidence of *EYS* dysfunction caused by G843E using zebrafish as a model.”

REVIEWERS' COMMENTS:

Reviewer #1 (Remarks to the Author):

The authors have successfully addressed all of my comments.

Reviewer #2 (Remarks to the Author):

I thank the authors for addressing each of my concerns in the initial review. I now have a better understanding of the manuscript and conclude that it contributes to the field by using GWAS in an innovative way to help determine pathogenicity of variants, which is currently a major problem in genetically diagnosing patients. My initial critiques have been adequately addressed. I would just like to point out an error in the legend of Table 3, in line 892- it should be "no" EYS path mut (they left out the no).

Reviewer #3 (Remarks to the Author):

The authors have addressed most comments on the original review. However, several comments need further attention. These are shown below. The original comment/author responses are shown. The remaining concern(s) is/are described below the (Author) response.

COMMENT #7:

3A: It appears that there is Eys staining also in the inner nuclear and inner plexiform layer.

29

This is also apparent in the sections from embryonic zfish retinas. Is this an expected location of Eys?

(Author) RESPONSE:

Thank you for the comment. You are right. As reported *ey* function is related cilia, so we focused on staining around the cilia, although ganglion cell layer and inner nuclear layer also showed some positive signal. However, the function of *ey* in inner retinal layers is completely unknown.

This comment needs to be further addressed. Expression of *ey* in the inner retina is clearly apparent, even stronger than what is shown in the photoreceptors. The expression of *ey* in the inner retina is briefly noted in the manuscript, when describing images presented in figure 4 (previously figure 3). However, the authors should also provide citations of papers where this expression has been previously described. Has this been observed in multiple species? In zebrafish? Is this work the first observation of such expression? A comment also indicating that function in inner retina is unknown should also be included.

(Author) RESPONSE:

Thank you for the comment. As you mentioned MOs cannot not suppress the gene expression for a long time. However, it can suppress the expression up until around 5 dpf (Neuron 42, 703-716, 2004). And this refers to gene expression but the induced phenotype could persist for a longer period of time. Second, we admit that we had not completely met the criteria. Thus, we added some experiments to further assess the efficacy of MO. The morphants in our experiment showed mislocalization of rhodopsin, a phenotype also observed in a zygotic mutant (knockout of *ey*) (Biol Open. 2016 15:1662-1673, Reference 29). Moreover, since co-injection of the wildtype *ey* mRNA had rescued the phenotype, we concluded that the morpholinos had given the *ey* knockdown-specific phenotype. Furthermore, we tested three different splice site morpholinos (SPMO1-3) in adult zebrafish. RT-PCR analyses showed inhibition of *ey* expression and immunohistochemistry displayed rhodopsin mislocalization with all MOs. Collectively, we believe that our experiment met the guideline that has been proposed by the reviewers.

The concern regarding 7 dpf sampling time after MO still remains. The reference information helped the authors' cause slightly but was not fully convincing. In that paper, from what I could tell, the MO analysis was primarily conducted at 3-4 dpf, and analyses at 5 dpf used a genetic mutant. Citation(s) for phenotype effects beyond 5 dpf, as claimed by the authors, need to be provided. If the authors wish to justify their sampling time, then citations for such a late sampling time after MO injection need to be provided for readers, when these experiments and data are presented (i.e. in the beginning of the Results section describing the MO experiments in zebrafish).

Why did the authors need to sample at 7 dpf for the rescue experiments, when they analyzed phenotype of *ey* morphants at 6 dpf? 6 dpf is still very late for MO analysis, but it could be considered better than 7 dpf.

The graph in Figure 4H needs y-axis label.

Line 231-232: Looking at the images in 4I-L, the phenotypes are not completely identical. This is also a blanket statement that would require examining more than only rhodopsin localization. It would be better stated that each morpholino resulted in mislocalization of rhodopsin in photoreceptors, which is said in the following sentence. Therefore, the sentence on line 231-232 should be removed.

The authors need to state the phenocopy of zygotic mutant, including the citation of the paper provided in the author response here, in the Results section in comparison to their findings with the anti-sense MOs.

COMMENT #10:

For the quantifications presented in Fig 3K, the sample size of n=5 or 6 is stated in the figure legend. How many cryosections were analyzed per embryo? At a minimum, 3-4 non-adjacent cryosections should be quantified. In addition, the sample size (actual n examined) would should be increased.

(Author) RESPONSE:

Thank you for pointing out an important issue. We usually arrange around 10 embryos per section and distinguish each other by their position on the section to avoid repeated counting of the same embryo. At least 6 non-adjacent sections per groups are assessed. In the revised manuscript, the sample size has been increased.

It appears that this data is now displayed in Figure 4Q in the revised manuscript: Is this quantification per eye (as stated in the figure legend) or per “field” (the graph’s y-axis label)? If it is “per field”, then what defines a “field”. The authors need to be very clear about how quantifications were performed, and what is being shown on the graph.

COMMENT #11:

What is the significance of rhodopsin mislocalization found in Fig 3G-J? Was this expected? Is this seen in RP or models of RP??

(Author) RESPONSE:

Rhodopsin mislocalization has been reported in mice models of RP (J Neurosci 1994, 14:5818, PNAS, 2002 99:5698., PNAS 2004, 101:16588). In addition, as *ey* is localized to the connecting cilium, the mislocalization of rhodopsin caused by defective ciliary transport is a reasonable phenotype. Moreover, we have demonstrated that the mislocalization can cause the photoreceptor cell death (PLoS One. 7, e32472, 2012).

This information, with citations, needs to be integrated into the manuscript. This can be done at the beginning of the Results section that describes the finding in zebrafish morphants.

Reviewer 1's comment

The authors have successfully addressed all of my comments.

Author RESPONSE:

Thank you very much for taking time to go over our work.

Reviewer 2's comment

I thank the authors for addressing each of my concerns in the initial review. I now have a better understanding of the manuscript and conclude that it contributes to the field by using GWAS in an innovative way to help determine pathogenicity of variants, which is currently a major problem in genetically diagnosing patients. My initial critiques have been adequately addressed. I would just like to point out an error in the legend of Table 3, in line 892- it should be "no" EYS path mut (they left out the no).

Author RESPONSE:

Thank you very much for finding interest in our work and pointing out an important error. The comment has been addressed accordingly.

Reviewer 3's comment

COMMENT #7:

3A: It appears that there is Eys staining also in the inner nuclear and inner plexiform layer. This is also apparent in the sections from embryonic zfish retinas. Is this an expected location of Eys?

(Author) RESPONSE:

Thank you for the comment. You are right. As reported eys function is related cilia, so we focused on staining around the cilia, although ganglion cell layer and inner nuclear layer also showed some positive signal. However, the function of eys in inner retinal layers is completely unknown.

This comment needs to be further addressed. Expression of eys in the inner retina is clearly apparent, even stronger than what is shown in the photoreceptors. The expression of eys in the inner retina is briefly noted in the manuscript, when describing images presented in figure 4 (previously figure 3). However, the authors should also provide citations of papers where this expression has been previously described. Has this been observed in multiple species? In zebrafish? Is this work the first observation of such expression? A comment also indicating that function in inner retina is unknown should also be included.

Author RESPONSE 2:

Thank you for the comment. There are not eyes in rodents, so it is difficult to discuss about it. However, recently, the expression of EYS in human eyes (Bonilha VL. et al. Graefes Arch Clin Exp Ophthalmol. 2015 253:295-305.) This article shows EYS expressed also in inner plexiform layer in human retina. We have cited the reference and altered the text

(Author) RESPONSE:

Thank you for the comment. As you mentioned MOs cannot not suppress the gene expression for a long time. However, it can suppress the expression up until around 5 dpf (Neuron 42, 703-716, 2004). And this refers to gene expression but the induced phenotype could persist for a longer period of time. Second, we admit that we had not completely met the criteria. Thus, we added some experiments to further assess the efficacy of MO. The morphants in our experiment showed mislocalization of rhodopsin, a phenotype also observed in a zygotic mutant (knockout of *eyes*) (Biol Open. 2016 15:1662-1673, Reference 29). Moreover, since co-injection of the wildtype *eyes* mRNA had rescued the phenotype, we concluded that the morpholinos had given the *eyes* knockdown-specific phenotype. Furthermore, we tested three different splice site morpholinos (SPMO1-3) in adult zebrafish. RT-PCR analyses showed inhibition of *eyes* expression and immunohistochemistry displayed rhodopsin mislocalization with all MOs. Collectively, we believe that our experiment met the guideline that has been proposed by the reviewers.

The concern regarding 7 dpf sampling time after MO still remains. The reference information helped the authors' cause slightly but was not fully convincing. In that paper, from what I could tell, the MO analysis was primarily conducted at 3-4 dpf, and analyses at 5 dpf used a genetic mutant. Citation(s) for phenotype effects beyond 5 dpf, as claimed by the authors, need to be provided. If the authors wish to justify their sampling time, then citations for such a late sampling time after MO injection need to be provided for readers, when these experiments and data are presented (i.e. in the beginning of the Results section describing the MO experiments in zebrafish). Why did the authors need to sample at 7 dpf for the rescue experiments, when they analyzed phenotype of *eyes* morphants at 6 dpf? 6 dpf is still very late for MO analysis, but it could be considered better than 7 dpf.

Author RESPONSE 2:

Thank you for the comment. We admit that it is not very common to analyze the effect of MO at 7 dpf. The reason why we used 7dpf embryo to analyze the phenotype is to facilitate the visualization of rhodopsin mislocalization. The mislocalization

phenotype was more readily visible in 7 dpf than in 6 dpf, the former providing reliable functional analysis of G843E.

However, the most important point is that we observed the same rhodopsin misvocalization phenotype induced by MO at 7 dpf as those induced by three different MOs analyzed at 6 dpf and reported in genetic mutants, which was clearly rescued by wildtype *ey3* mRNA but not by mRNA with G843E mutation. This should indicate the pathogenicity of G843E mutation.

Furthermore, as we mentioned earlier, induced phenotype could persist beyond the period of knockdown. Thus, logically, the results are not unexpected

The graph in Figure 4H needs y-axis label.

Author RESPONSE 2:

Thank you, this has been addressed

Line 231-232: Looking at the images in 4I-L, the phenotypes not are not completely identical. This is also a blanket statement that would require examining more than only rhodopsin localization. It would be better stated that each morpholino resulted in mislocalization of rhodopsin in photoreceptors, which is said in the following sentence. Therefore, the sentence on line 231-232 should be removed.

Author RESPONSE 2:

We have followed the suggestion and corrected the text.

The authors need to state the phenocopy of zygotic mutant, including the citation of the paper provided in the author response here, in the Results section in comparison to their findings with the anti-sense MOs.

Author RESPONSE 2:

Thank you. We have altered the text.

COMMENT #10:

For the quantifications presented in Fig 3K, the sample size of n=5 or 6 is stated in the figure legend. How many cryosections were analyzed per embryo? At a minimum, 3-4 non-adjacent cryosections should be quantified. In addition, the sample size (actual n examined) would

should be increased.

(Author) RESPONSE:

Thank you for pointing out an important issue. We usually arrange around 10 embryos per section and distinguish each other by their position on the section to avoid repeated counting of the same embryo. At least 6 non-adjacent sections per groups are assessed. In the revised manuscript, the sample size has been increased.

It appears that this data is now displayed in Figure 4Q in the revised manuscript: Is this quantification per eye (as stated in the figure legend) or per “field” (the graph’s y-axis label)? If it is “per field”, then what defines a “field”. The authors need to be very clear about how quantifications were performed, and what is being shown on the graph.

Author RESPONSE 2:

Thank you for the comment. The number of cells per retinal section was quantified. We made this clear in the Method section and the figure legend.

COMMENT #11:

What is the significance of rhodopsin mislocalization found in Fig 3G-J? Was this expected? Is this seen in RP or models of RP??

(Author) RESPONSE:

Rhodopsin mislocalization has been reported in mice models of RP (J Neurosci 1994, 14:5818, PNAS, 2002 99:5698., PNAS 2004, 101:16588). In addition, as *eyes* is localized to the connecting cilium, the mislocalization of rhodopsin caused by defective ciliary transport is a reasonable phenotype. Moreover, we have demonstrated that the mislocalization can cause the photoreceptor cell death (PLoS One. 7, e32472, 2012).

This information, with citations, needs to be integrated into the manuscript. This can be done at the beginning of the Results section that describes the finding in zebrafish morphants.

Author RESPONSE 2:

Thank you. I have altered the text accordingly.